# Real-time encoding and compression of neuronal spikes by metal-oxide memristors

Isha Gupta[1,*], Alexantrou Serb[1,*], Ali Khiat[1], Ralf Zeitler[2], Stefano Vassanelli[3] & Themistoklis Prodromakis[1]

Advanced brain-chip interfaces with numerous recording sites bear great potential for investigation of neuroprosthetic applications. The bottleneck towards achieving an efficient bio-electronic link is the real-time processing of neuronal signals, which imposes excessive requirements on bandwidth, energy and computation capacity. Here we present a unique concept where the intrinsic properties of memristive devices are exploited to compress information on neural spikes in real-time. We demonstrate that the inherent voltage thresholds of metal-oxide memristors can be used for discriminating recorded spiking events from background activity and without resorting to computationally heavy off-line processing. We prove that information on spike amplitude and frequency can be transduced and stored in single devices as non-volatile resistive state transitions. Finally, we show that a memristive device array allows for efficient data compression of signals recorded by a multi-electrode array, demonstrating the technology's potential for building scalable, yet energy-efficient on-node processors for brain-chip interfaces.

[1] Department of Electronics and Computer Science, Faculty of Physical Science and Engineering, University of Southampton, University Road, SO17 1BJ Southampton, United Kingdom. [2] Max Planck Institute for Intelligent Systems, Heisenbergstr, 3, 70569 Stuttgart, Germany. [3] Department of Biomedical Sciences, University of Padova, Via Francesco Marzolo 3, Padova 35131, Italy. * These authors contributed equally to this work. Correspondence and requests for materials should be addressed to I.G. (email: I.gupta@soton.ac.uk).

Understanding brain function relies heavily upon long-term recording of neuronal populations. Advances in chip-based neuronal probe technology[1–4] have led to recording systems capable of monitoring in real-time large numbers of neurons[5,6], bearing great potential for fundamental neuroscience and neuroprosthetic applications[7–9]. Currently, probes with multi-electrode-arrays (MEAs) are capable of simultaneously recording electrical activity from up to 512 sites at 40 k samples per second *in vivo*[10] and from up to 32,768 sites at 2.4 k frames per second *in vitro*[11]. Further advances, particularly towards fully implantable autonomous systems, are hindered by real-time processing of the streamed neuronal signals, which would notably increase power dissipation along with dropping of signal-to-noise ratio (SNR)[12]. Addressing these challenges necessitates the intelligent compression of big neural data generated[13] via on-node processing, currently pursued by shifting the spike detection and sorting task on-chip via template matching[6,14,15]. However, the resulting scalability issues, the drive for even further power budget reductions together with the consideration that neuroprosthetics have been successfully operated with simple, rate- or spike-count-coded input signals[16–20] have kindled interest in processing neuronal signals in a bio-inspired fashion. This justifies current interest in leveraging emerging technologies for resurrecting Carver Mead's original vision in neuromorphic systems[21], where efficient data processing is implemented for example through artificial retinas[22].

Memristive devices appear to be well suited in providing a disruptive technological boost to this vision by performing the role of artificial synapses. Much akin to biological synapses, they possess the intrinsic ability to simultaneously carry out computational tasks and store information at aggressively downscaled volumes and power dissipation[23,24]. Here we exploit the intrinsic characteristics of metal-oxide TiO$_x$-based memristors, such as their analogue memory capacity that occurs above certain voltage thresholds for encoding and compressing neuronal spiking activity recorded by MEAs. We demonstrate how a large part of the computational burden associated with spike detection can be relegated to single memristive devices that can be accommodated in the back-end-of-line of complementary metal-oxide semiconductor (CMOS) technologies, along with neuronal probe manufacturing.

## Results

**Memristors as events integrators**. As originally proposed by Chua, memristors are capable of changing their resistive state as a function of the integral of their input voltage; a phenomenon known as resistive switching[25]. As a result of this single-device integrator property, solid-state implementations of memristive devices[26–28] have been at the center of attention, with potential applications in emerging memories and neuro-inspired computing[29]. In this work, we exploit metal-oxide-based resistive switches as neuronal spike integrators. Solid-state TiO$_x$ memristors with a metal-insulator-metal architecture, as shown in Fig. 1a, were fabricated on a Si/SiO$_2$ substrate; detailed process parameters appear under the Methods section. Subjecting the device-under-test (DUT) to a train of input programming pulses in alternating polarities gives rise to gradual resistive state transitions, provided the pulse amplitude exceeds the device's inherent bipolar switching thresholds (fundamental properties of the device; denoted as $V_{th-}/V_{th+}$), as illustrated in Fig. 1b,c (see Methods). Here we argue that this capability for gradual switching can be exploited to encode multiple significant spiking events as small changes in a device's resistive state. This assumption is first explored deterministically, by employing known pulse events. Figure 1d,e show the response of a typical

DUT to trains of 200 identical square-wave events of negative and positive polarities[30], respectively, as illustrated in the insets. Each writing pulse has a fixed 100 μs duration and suitable amplitude to induce a resistive state change. It is followed by a reading pulse of fixed 0.5 V amplitude and an automatically determined duration, $t_a$ (ref. 31). Notably, the pulse amplitude required to elicit a resistive state change of similar strength but in the opposite direction could differ, indicating an inherent asymmetry in the device's characteristics. This bidirectional, gradual (analogue), saturating switching, could be fitted by second order exponential functions of input voltage integral (Supplementary Table 1), thus defining the input-output relation of an integrating sensor for distinct stimulation protocols. Notably, as our TiO$_x$ device prototype acts as a thresholded integrator, it can be described by the generalized definition of memristor as 'zero phase-shift dynamic system'[32]. We name hereafter the device as memristive integrating sensor (MIS) and show that this thresholded-integrator attribute can be particularly useful for compressing information and suppressing noise in signals with low SNR, such as data recorded from the activity of neurons/cells. Hence, our approach only allows for significant, supra-threshold events to be registered as measurable changes in the device memory state, whilst sub-threshold events are suppressed.

**Neural spiking integration with metal-oxide memristors**. The ability of memristors to integrate significant events provides an efficient way of encoding and compressing information on neuronal firing in real-time, as recorded by neuronal probes. The basic concept of the proposed MIS platform is exemplified in Fig. 2a. An external front-end platform (for example a MEA) senses neuronal electrical activity which is fed into the MIS system as a series of voltage-time samples. The MIS begins by pre-amplifying the incoming signal to voltage levels suitable for operating the memristor sitting at the core of the MIS and then proceeding to apply the pre-amplified signals to the memristor in real-time. Periodically, the memristor's resistive state is assessed periodically and when a significant change in comparison to the previous state is detected, the system registers a spiking event.

We validated experimentally the MIS system implementation on spiking activity of retinal ganglion cells. At first, the activity of dissected retinal cells was pre-recorded by an external MEA front-end system[2,33–36] (see Methods, CMOS MEA). The MEA employed records the raw bio-signals, which lie in the 0.1 mV–1 mV range, and then uses its own in-built amplifiers to boost them to the 10 mV–100 mV range. The resulting, boosted recordings are then stored off-line as voltage-time series. For this work, we have used these stored recordings as inputs to our MIS platform in isolation from the front-end, that is the front-end has not been connected to the MIS platform in real-time (Supplementary Fig. 3). The processing of neural signals through MIS platform begins when the stored voltage-time series are subjected to amplification and offset in software on the PC that runs the platform (Fig. 2a—box (i) and Methods section). This set-up offers the option of adjusting the MIS detection threshold and consequently allowing the integration of significant spiking events with a pre-determined SNR. For example in Fig. 2c, the offset and scaling parameters were chosen such that only the most significant events (largest amplitude extracellular spikes) would exceed the threshold. The resulting, pre-conditioned waveform is then transmitted from the PC to the memristor testing and operation instrument (see Methods, Hardware Infrastructure), which physically implements the MIS system. The instrument, in turn, plays back the waveform to a target memristive device.

In order to assess the distinct resistive state changes during the streaming of recorded neural activity (Fig. 2b—see Methods),

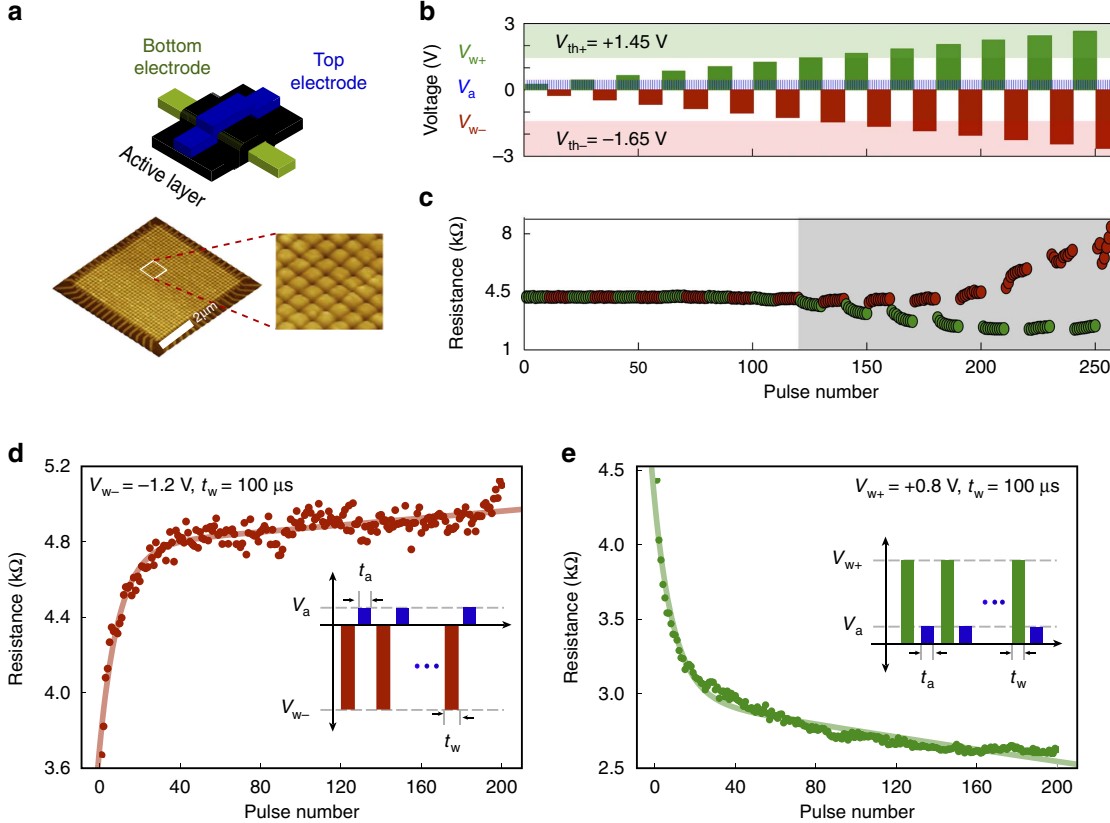

**Figure 1 | Device architecture and electrical characterization of solid-state TiO$_x$ resistive random access memory devices. (a)** Schematic illustration of a solid-state TiOx memristive device and atomic force microscopic (AFM) image of $32 \times 32$ crossbar array. (**b,c**) Resistive state changes (bottom trace) accumulate visibly, and in opposite direction depending on polarity, only in response to input pulses with above-threshold amplitudes (top trace; input writing pulses, $V_{W+}$ and $V_{W-}$, indicated in red and reading pulses of amplitude $V_a$ in light blue). In bipolar devices two inherent thresholds exist, one for each voltage polarity. For this device we obtained $V_{th+} = +1.45$ V and $V_{th-} = -1.65$ V as indicated by the shadowed areas of the plot. (**d,e**) Show gradual resistive switching under a pulse train stimulation (200 pulses per train). The devices response is fitted with a second-order exponential function (continuous line). Typical biasing scheme parameters (insets): negative write pulse voltage $V_{w-} = -1.2$ V, positive write pulse voltage $V_{w+} = +0.8$ V, read pulse voltage $V_a = 0.5$ V, write pulse width $t_w = 100$ µs and read pulse width $t_a$ automatically determined by the measurement system.

the DUT is periodically disconnected from the neural signal feed, for example once every 200 input samples, and connected to a read-out circuit that captures the device's state, digitizes it and subsequently sends the resistive state reading back to the PC. Importantly, only a limited amount of data is returned to the PC when compared with the full voltage time series in conventional systems (box (ii) in Fig. 2a and see Methods, MEA neural recording signal-processing). A software converts the incoming series of resistive state readings into a series of resistive state changes, subsequently keeping only the largest ones that are marking significant events in the neural signal (Fig. 2a marked (iii), Methods and Supplementary Fig. 6). Noteworthy this filtering process, based on an assessment of resistive state changes in absentia of an input signal (see Methods section) may be engineered in order to fine-tune SNR on neuronal activity.

Our effort to reduce data bandwidth echo current research in on-chip spike-sorting[8], with our approach being disruptive in exploiting the inherent data-compression capability of highly scalable, low-power nanodevices that could extend the scaling and processing capacity of neural recording platforms substantially. Our approach reproduces in its essence the strategy adopted by natural synapses for signal compression, where information on spikes number and firing rate is stored into gradual changes of the postsynaptic membrane conductance. In contrast, present state-of-art neural activity monitoring platforms, like the MEA-based system described in ref. 33, rely entirely on

front-end circuitry for detecting and transmitting all data offline for processing (Supplementary Fig. 7).

**MIS system performance**. MIS system performance was investigated in three separate experiments. First, the capability of handling input signals where neuronal spikes span both negative and positive voltages was tested including a repeatability check. Second, the spike detection performance was benchmarked against a state-of-art template-matching system[22]. Finally, robustness checks were carried-out.

In the case of *in vivo* recording spikes often span both negative and positive voltage polarities, depending on experimental conditions, for example the position of the recording electrode relative to neuronal compartments and their associated ionic conductances[5,37]. It is thus relevant to demonstrate the MIS operation for both signal polarities, as explored here at a proof-of-concept level. Figure 2c depicts a waveform consisting of four, concatenated copies of a retinal recording. Each copy was subjected to appropriate scaling and offsetting, and two of the copies were polarity-inverted. The corresponding resistive state transient response throughout this test is shown in Fig. 2d. Significant changes in resistive state correspond to clear, supra-threshold events. We demonstrate that spike detection successfully occurs at both polarities, achieving qualitatively similar modulation over two signals. This is better illustrated in Fig. 2e, where the normalized resistive state changes between

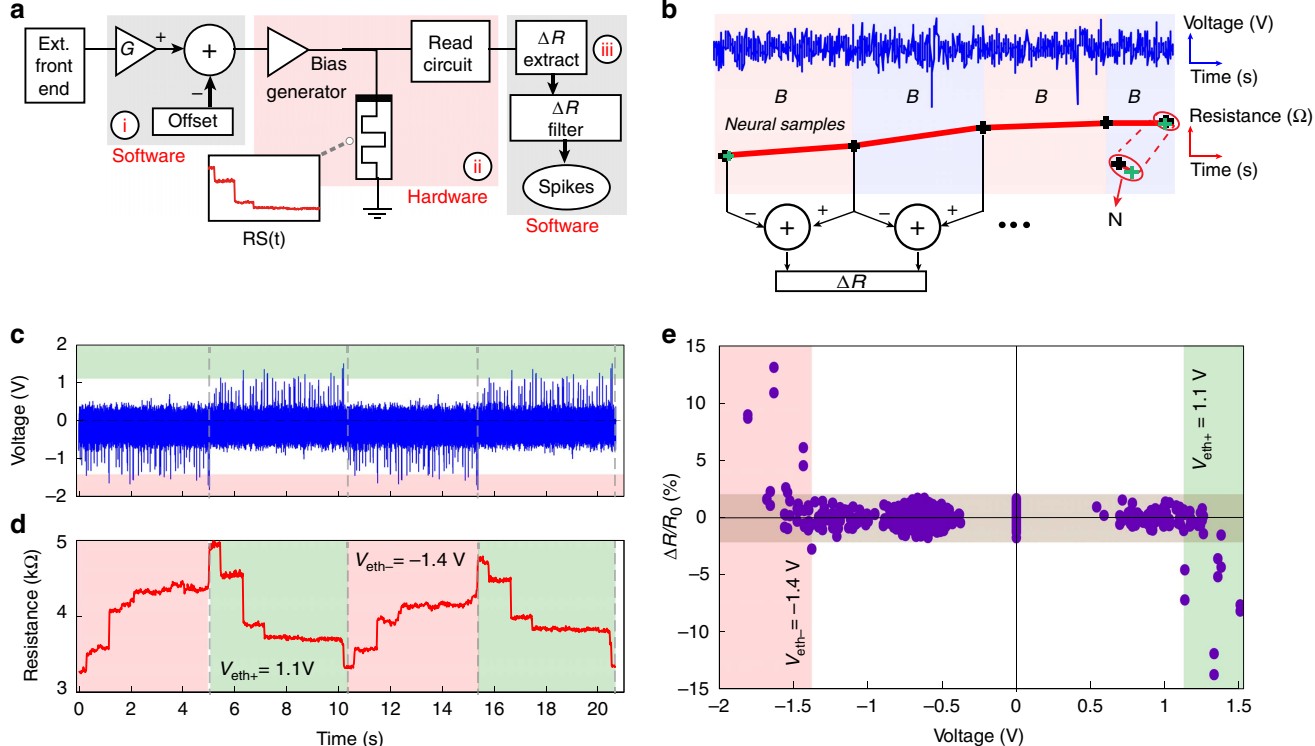

**Figure 2 | MIS concept and operation. (a)** Block diagram of the signal processing in the proposed spike-detection system. An external frontend (a CMOS MEA system) located externally to the MIS platform records extracellular neuronal signals and amplifies them. The pre-amplified, acquired neural recordings are then fed into our instrument, suitably gain-boosted ($G$) and offset ($V_{off}$) to render them compatible with the memristors' voltage operating regimes (i). The conditioned waveform is fed into a memristor and its resistive state is then periodically assessed (ii). Changes in resistive state caused by spiking events are extracted offline (iii). **(b)** Conceptual read-out scheme for evaluating the time evolution of the resistive state of test devices subjected to input stimulation for one batch. The resistive state (red line) is assessed at the beginning of each neural recording batch (blue trace), then every chosen number of samples termed as bin ($B$) and finally at the end of each batch (assessment points marked by crosses). Changes in test device resistive state ($\Delta R$) are extracted from consecutive resistive state assessments. Resistive state changes occurring between the last measurement of each batch and the first measurement of the next batch, with no interceding pulse biasing, ($N$) are considered to result from measurement uncertainty and can be used to determine the noise band. **(c,d)** Shows an arbitrary input waveform consisting of four concatenated copies of the same retinal cell recording and artificially inverted to produce spike trains with alternating polarities. This waveform was employed to validate the concept of memristive integrating sensors, the response of which is shown in **d**. The collated recording copies in **c** have been subjected to appropriate scaling and offsetting in order to accommodate the device's asymmetric threshold voltages, resulting in balanced resistive state SET and RESET. The extracted threshold voltages are identified here as, $V_{eth+} = 1.1$ V and $V_{eth-} = -1.4$ V represented in the green and pink band, respectively; x axis for both (**c,d**) is given in S.I. units—each data sample lasts 82 μs (sampling frequency: 12.2 kHz). **(e)** Fractional resistive state modulation ($\Delta R/R_O$) extrapolated from (**d**) showcasing significant resistive state modulation occurring only above $V_{eth+}$ and below $V_{eth-}$ while intermediate bias values (noise) leads to no significant change.

consecutive reads is plotted as a function of the maximum voltage magnitude of interceding events. In the same figure, the grey horizontal band denotes resistive state changes that have been discarded (see Methods section). The remaining points are used to define the memristor's effective operating threshold voltages ($V_{eth+/-}$), which partition the plot into three distinct areas: two of them correspond to significant resistive state modulation (larger than $V_{eth-}$ and less than $V_{eth+}$) and the last one ($[V_{eth+}, V_{eth-}]$) containing resistive state changes that are indistinguishable from the estimated background noise. The range of effective threshold voltages for the TiO$_x$ prototypes employed throughout this study was $-0.8$ V to $-1.8$ V (Supplementary Fig. 2). Importantly, whilst the inherent threshold of the device performs a coarse filtering action, the effective threshold ultimately determines SNR. Moreover, since the MIS system detects normalized changes in the resistive state, this approach is inherently robust against the devices threshold variability as identified in Supplementary Fig. 2b.

The performance of the introduced MIS concept was benchmarked against a state-of-art template-matching-based system[22] (Supplementary Fig. 7). The resulting performance

comparison between the two approaches is presented in Fig. 3. In this case, we employed an offset ($V_{off} = 0$) and amplification ($G = 2.8$) on the recording shown in Fig. 3a and the device's resistive state was assessed as per the standard scheme described in the Methods section and in more detail in Supplementary Fig. 5. Figure 3b illustrates the resistive state evolution of the tested MIS in response to the input signal shown in Fig. 3a. One can observe clear changes in the device's resistance corresponding to spiking events whose magnitude exceeds $V_{eth-}$, in a similar manner to the first period of events shown in Fig. 2c,d. In this example, the incoming spikes mainly occur in negative polarity hence there is an overall inscrease in resistance, from approximately 2.5 to 5.5 kΩ. However, the presence of a few events in opposite polarity that exceed $V_{eth+}$, cause occasional resistive state drops. A clear example indicated by φ in Fig. 3g,h can be observed at ~1.4 s in Fig. 3a,b,e,f where the resistive state reduces from ~4.5 to 4 kΩ. However, optimizing the value of $V_{off}$ provides additional flexibility for compensating for this effect. Noteworthy, as the MIS is capturing and storing significant events as non-volatile resistive changes, one can afford to use relatively low sampling rates for minimizing the overall requirements in

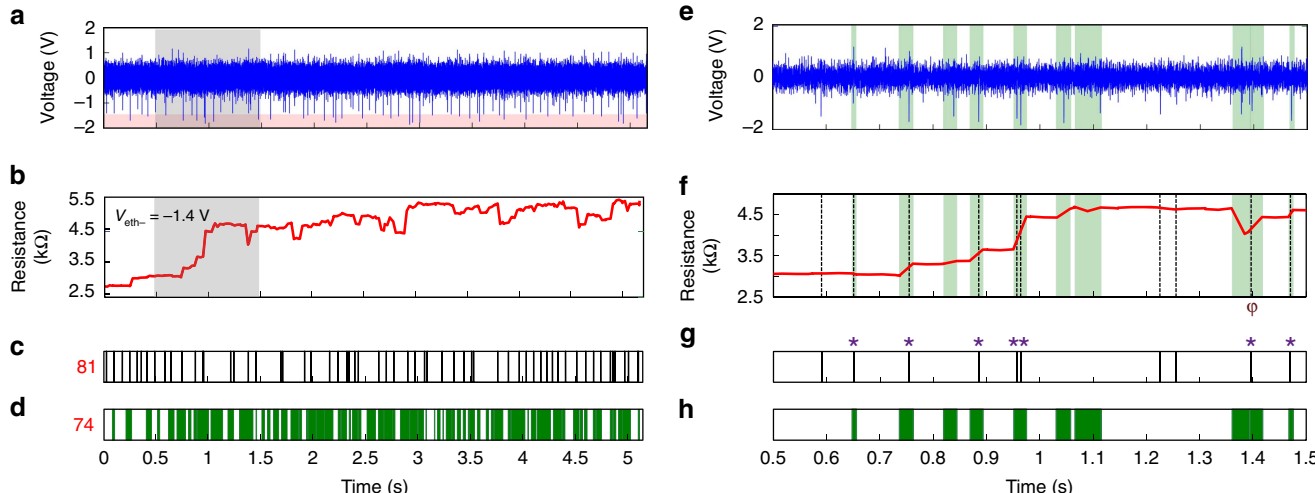

**Figure 3 | Benchmarking memristor-based system against state-of-art template matching system.** (**a**) A pre-conditioned neural recording trace with gain and offset value of 2.8 and 0, respectively, causes the resistive state time evolution shown in **b**. (**c**) Eighty-one spikes were detected by the template matching system, with grey lines indicating spike positions. (**d**) Green shading indicates time intervals within which one or more spikes were detected through the MIS; total of 74. (**e,f**) are close-ups of the neural recording and resistive state evolution shaded grey in **a,b**, respectively. Time intervals where the MIS detects spikes are shaded green while the locations of spikes detected by the template matching system are indicated by grey vertical dashed lines. (**g,h**) Comparison of the detection of spikes by the two systems. The asterisk mark (*) indicates agreement between the two systems and the φ symbol indicates the resistive state drop associated to the occurrence of a large-amplitude positive event.

data storage/handling. Along this line, the output of our system is quantified at discrete time bins containing one or more detected events (see for example Fig. 3e, f–h at ∼0.96 s). For this recording, our system identified 74 bins containing significant events denoted in Fig. 3d, while the template-matching-based system overall distinguished 81 significant events shown in Fig. 3c. Comparing the MIS and template-matching approaches within a representative time-window of 1 s duration, indicates a similar performance in spike detection, as noted via asterisk symbol (*) marks in Fig. 3g,h. It is interesting to note that our approach results into registering apparent events (for example at 0.83, 1.05, 1.1 s) that are missed by the template-matching method while it fails recognizing other possible events (for example at 0.59, 1.22, 1.25 s), presumably due to a conservative selection of signal conditioning gain. Overall, benchmarking the efficiency of the MIS approach indicates a rate of true positives of ∼60%, (Supplementary Fig. 6) assuming that the template-matching approach is an ideal spike detector.

The robustness of the observed behaviour and its potential for data compression via improvement of SNR is further demonstrated by showcasing: (a) the response of a single MIS to blocks of neural recording data containing significantly different patterns of activity (Fig. 4a–d) and (b) the response of different devices to a common neural recording obtained from MEA as exemplified in Fig. 4e–h. In the former one device—many recordings case we observe how intense activity leads to a larger overall resistive state modulation and how particularly strong events tend to cause distinct non-volatile changes in memory states. Thus, resistive state traces compress information on both the firing rate and spikes amplitude. Instead, in the latter one recording—many devices case we observe that despite the quantitative variability in device behaviour, most of the marked resistive state transitions tend to concur in time with significant events present in the input waveform (see also Supplementary Table 2).

**Towards array-level MIS operation**. The concept introduced in Fig. 2a, when directly interfaced with front-end-circuitry, can be exploited for advancing the present state-of-art in high-density

neural recording platforms[38]. The presented concept is amenable for scaling to a multi-channel array level, as illustrated in Fig. 5a, for capturing the activity of neural networks in real-time. We envisage an overall system architecture very similar to standard active pixel sensor CMOS imagers[39]. In this hybrid system, data from each of the N pixels in the array arrives as an analogue current from the MEA and is multiplexed onto one of the M on-chip trans-impedance amplifier (TIA) blocks, which are followed by on-chip offset stages. Thus, a small number of both gain and offset stages are time-shared by every pixel in the array. The conditioned recording data points are then de-multiplexed to a memristor bank, that can be integrated into the back-end of the chip, in good proximity to the MEA recording sites. MIS output is then generated by sequentially measuring the resistive states of each memristor in the bank. The low frequency at which memristor read-outs are generated (for example 200 times lower data rate vis-à-vis input stream arriving from the MEA if a standard scheme is used as described in Methods section) allows the MIS system to carry out all measurements through a single or few, time-shared TIA feeding into analogue-to-digital converter. The digitized results are then sent off-chip. We foresee that, a practical implementation of a monolithically integrated system will involve addressing the challenges associated with the integration of a MIS array with CMOS-based front-end circuitry, while the required MIS control can be accommodated as peripheral circuitry with sneak-path issues existing in dense resistive random access memories crossbar configuration mitigated via selector topologies[40].

In this work, this concept was validated via a hybrid approach that is capable of processing 224 distinct recording traces stemming from a 16 × 14 pixel subset of the previously employed MEA system[2,34], atop which retinal ganglion cells were cultured. The sub-array was found to cover three cells after processing all recordings with a state-of-art array-level template matching system, using an extended principal component analysis method (described in Supplementary Fig. 8). As before, an initial MIS calibration was performed in order to set suitable values for G and $V_{off}$. This entailed selecting a spatially sparse subset of 23 pixels (see Supplementary Fig. 9, cells marked in orange), and

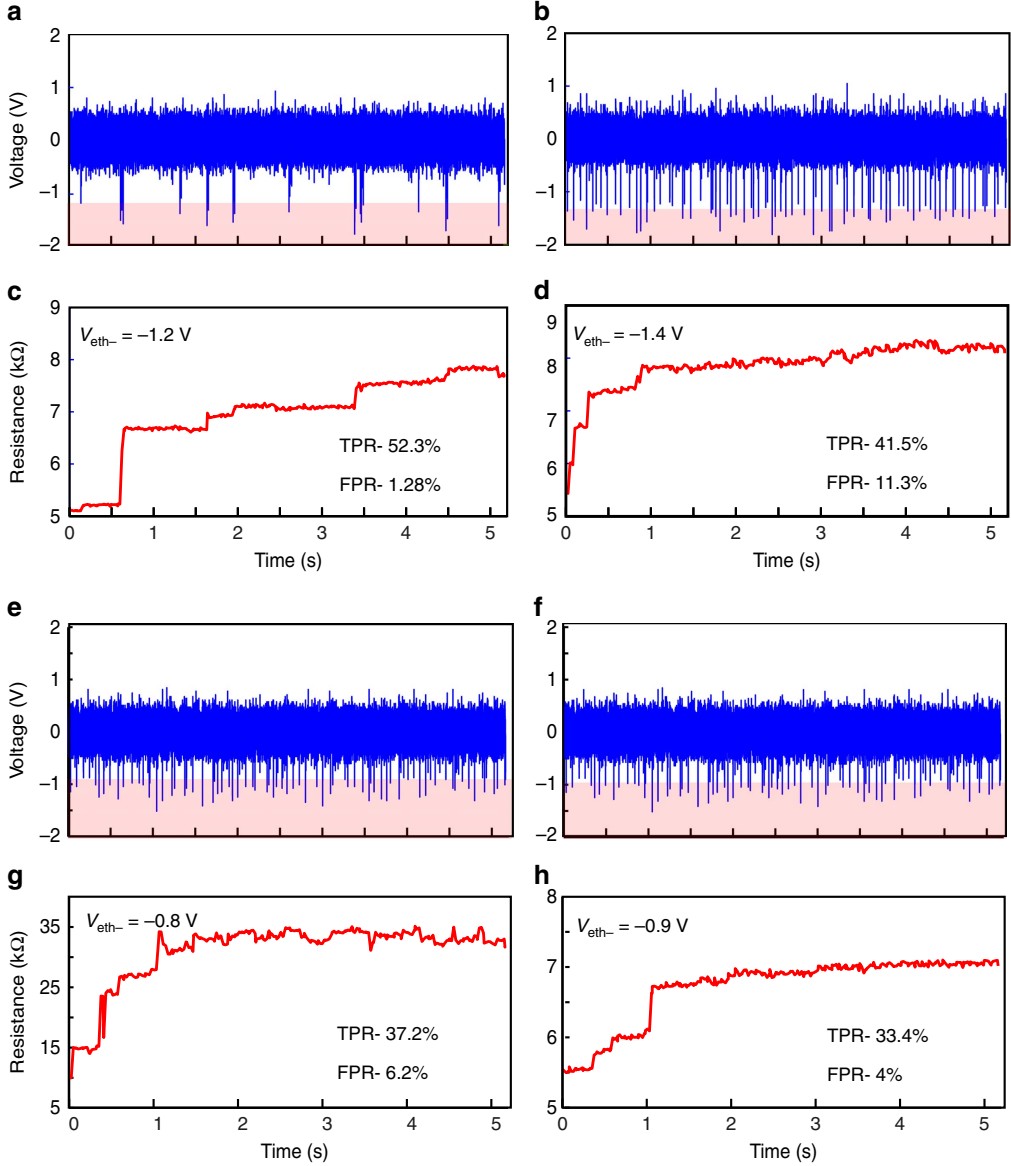

**Figure 4 | Robustness of memristive devices.** (**a–d**) Response of a single memristor (**c,d**) to two blocks of neural recording data (**a,b**) containing significantly different patterns of activity. The pink band indicates the extracted threshold of the device-under-test ($V_{eth-}$). The respective figures indicate quantification parameters, that is rate of true positives (TPR) and false positives (FPR), respectively. (**e–h**) Response of two different devices (**g,h**) to a common reference block of neural recording data represented in **e,f**. Initial resistance of device was set to $\sim 5\,k\Omega$. In all these experiments, signal conditioning parameters, that is software added gain and offset remained fixed at $G = 2.2$, $V_{off} = 0$, respectively.

examining their recording waveforms to gauge average maximum/minimum voltage amplitudes as well as the typical levels of background activity. In this experiment, we particularly set $G = 2.8$ and $V_{off} = 0$ for all memristive devices to ensure a suitable SNR. These parameters were kept fixed for all recordings, they were not changed for accommodating individual memristive device behavioural variations, or distinct features of the employed recordings. Every utilized memristive device was initialized to a common low-resistive state (Supplementary Fig. 1) in the range of $2$–$4\,k\Omega$ that for the given parameters yielded a useful MIS operating range up to the set $15\,k\Omega$ high-resistive state.

We further monitored the spatio-temporal changes in the array's memory state, snapshots of which are shown in Fig. 5b–d for distinct time instances: $t_1 = 1.63\,s$, $t_2 = 3.27\,s$ and $t_3 = 5.16\,s$ respectively. Since the neuronal activity is encoded as non-volatile resistive state changes we were able to observe an accumulation of activity clustered around three major centres: at pixel (row,

column) locations (3,4), (7,10) and (11,7). Particularly the final array state, shown in Fig. 5d, qualitatively resembles the activity extracted by the conventional template matching method to the same neural recording data set as shown in Supplementary Fig. 8. We note that while the system in Supplementary Fig. 8 outputs a spike count that is insensitive to the amplitude of the detected spikes, the proposed MIS array results into a ratiometric change in resistive state that is strongly correlated to the strength of the individual spiking events. This allows us to preserve information on both event amplitude and polarity, which in principle improves the data compression rate. We also note a few pixels exhibiting strong resistive state changes despite not appearing to belong to any well-defined cluster of activity (see Supplementary Fig. 10). This discrepancy follows the argument presented previously for Fig. 3g,h, hinting that single, exceedingly strong events may lead to resistive state changes comparable to those arising as a result of accumulated activity.

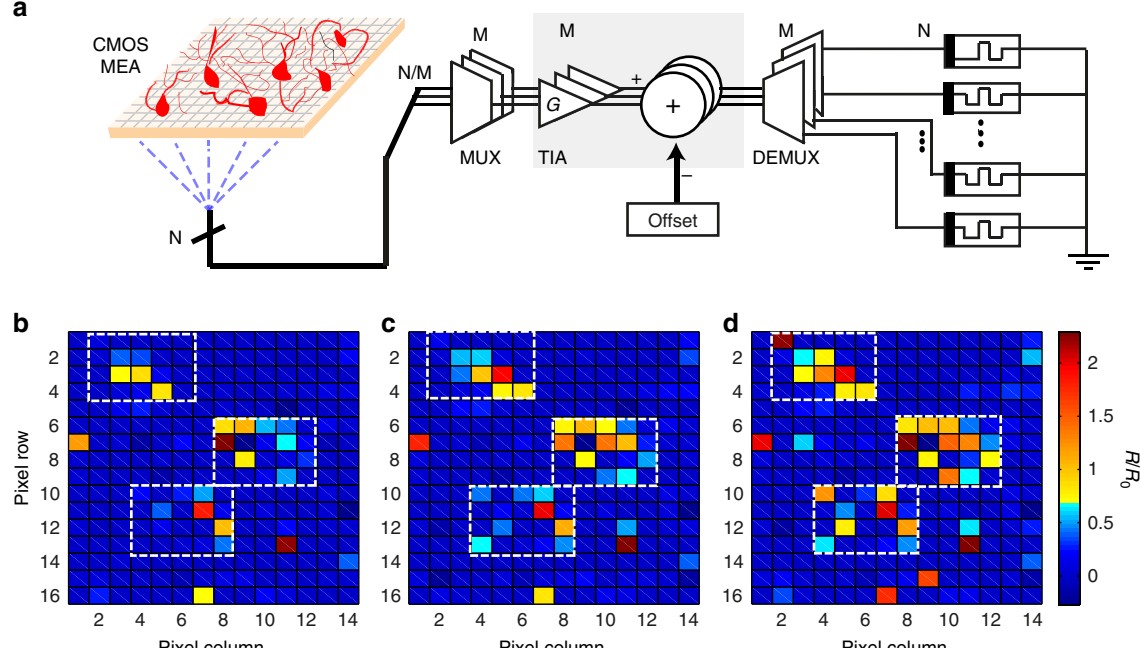

**Figure 5 | Towards array level integration.** (**a**) Conceptual diagram indicating conditioning of data from N pixels through multiple gain and offset cascade (M). (**b–d**) Time evolution of normalized resistive state throughout $16 \times 14$ test sub-array at $t_1 = 1.63$ s, $t_2 = 3.27$ s and $t_3 = 5.16$ s, respectively. The gain and offset values for the neural recordings was fixed at 2.8 and 0, respectively. Three clusters of activity can be discriminated. CMOS MEA, multi-transistor array block, manufactured in standard, commercially available CMOS technology. TIA, trans-impedance amplifier converting current to voltage with appropriate amplification.

## Discussion

Future autonomous and fully implantable neuroprosthetic platforms will have to rely on innovative strategies for low-power on-chip processing of neuronal signals. In neuron-to-neuron communication, information carried by spikes is effectively compressed in changes of synaptic strength. In a similar fashion, information of spikes recorded by neural implants can be stored, in a compressed form and with minimal power consumption, via single memristive devices. In this work, we demonstrated this concept that a memristor-based neural activity sensor could operate as a neuronal spike encoder, by compressing information on the spikes amplitude and firing rate. By extending the idea to the array level, we demonstrate that our approach is potentially suitable for monitoring the activity of multiple cells at large-scales.

The required power budget of our approach can be optimized by mapping the highest amplitude neural recording samples onto $\sim 5$ V pulses at $< 100$ ns due to the known voltage-time trade-off[41] (see Supplementary Fig. 11). Under the realistic assumption of operating devices at resistive states of $100$ kΩ, every 1,000 samples (one data batch, as per the standard schematic) we would spend a maximum of $250 \,\mu$W multiplied by $100$ ns $= 25$ pJ for biasing a device with neural recording data. Simultaneously, the read-out operation would cost $0.8 \,\mu$W multiplied by $100 \,\mu$s $= 80$ pJ (based on $0.4$ V read out voltage), rendering an average power dissipation of approximately $300$ nW per channel. Clearly, the memristor read-out and biasing circuitry will require an additional power. Nevertheless, estimated figures are already significantly less than the present state-of-art continuous time spike-detectors[15]. Most importantly, the proposed technology is demonstrated here at a proof of concept-level via large prototype devices and clearly the presented power/density considerations are not a reflection of the technology's full potential. We can expect that substantial improvements in power consumption can be achieved by further

downscaling and/or operating memristors at even higher resistive state ranges, for example operating the device in $1$ MΩ region can further reduce the power dissipation remarkably by two orders of magnitude. In addition, the bandwidth required to assess the resistive state should be rather low as compared with the raw input data-rate. Additional power efficiency gains can be expected by integrating the MIS elements atop state-of-art CMOS thus minimizing parasitic capacitances.

Finally, the key focus for driving this work forward in the future is to improve on the detection accuracy rates. A plausible performance-limiting factor is the programming saturation of memristive devices. This can however be counteracted by optimizing the main operational parameters that is gain and offset settings (Supplementary Fig. 12) or via employing memory state resets, as depicted in Supplementary Fig. 13. Another possible line of investigation towards MIS-based spike-sorting lies in determining how much information on spike amplitude/ duration can be extracted from the history-dependent magnitude of resistive state changes. In conclusion, the introduced MIS concept shows real promise for advancing and complementing the current state-of-art neural recording systems towards improving the power and area requirements of emerging bioelectronics.

We have demonstrated a novel recording system concept exploiting the intrinsic synapse-like attributes of metal-oxide memristive devices to compress information on neuronal firing. Our results show that single devices are capable of identifying significant spiking events while suppressing noise, thus paving the way towards highly area- and energy-efficient on-node neural recording processing. Contrary to time-domain sampling, the proposed MIS encode the presence of events in non-volatile resistive state changes, allowing the flexibility to trade off sampling rates for timing resolution. This is particularly useful when information is rate- or spike-count-coded and where only a measure of overall activity within given time bins is requested.

Typically, this is the case for brain-chip interfaces and neuroprostheses, where power dissipation linked to processing remains a major challenge. Moreover, as the memristor resistive state changes are linked to amplitude and polarity of the input waveform (signal envelope) this information is preserved in the magnitude of resistive state modulation. Finally, we note that this concept can be generalized for enabling smart data compression in distinct sensing platforms, particularly relevant to pervasive sensing systems.

## Methods

**Fabrication.** All the devices exploited in this work were fabricated according the following flowchart; 200 nm of insulating $SiO_2$ was thermally grown on 6-inch Silicon wafer. Then three main patterning steps were processed, each contains optical lithography, film deposition and lift-off process. In the first step, 5 nm Titanium (Ti) and 10 nm Platinum (Pt) films were deposited via electron-beam evaporation technology to serve as bottom electrodes, Ti was used for adhesion purposes. In the second, magnetron reactive sputtering system was used to deposit the $TiO_x$ ($x = 0.06$) active core from Ti metal target. Two plasma sources were used to ensure near stoichiometric film. 25 nm thick $TiO_x$ was deposited. In the final step, 10 nm Pt top electrodes were deposited using electron-beam evaporation system. At the end of processing, the wafer was diced into 9 by 9 $mm^2$ chips, which were then wire-bonded in standard packages for measurements and 60 by 60 $\mu m^2$ devices were used for the experiments.

**Device characterization.** The $TiO_x$ devices initially undergo an electroforming step[42] (inset of Supplementary Fig. 1). A voltage sweep is applied on a pristine sample until a sudden, non-volatile memory transition to the ON (low resistive state) state is observed. This typically occurs at $\sim +6.5$ V. Thereafter the device enters its normal operating regime, where it supports reversible resistive switching. Notably in such regime, and similarly many families of practical resistive random access memories, the intrinsic voltage threshold accounts for the response to voltage pulsing events. This memristive behaviour is apparent in Fig. 1b,c where a DUT was subjected to trains of input programming pulses in alternating polarities at a fixed duration 100 µs (write operation). The device memory state was read after each programming pulse at approximately 0.5 V. Significant changes in resistive state are observed, that is switching of devices to high-resistive state (RESET) and low-resistive state (SET) with negative and positive polarity, respectively, in Fig. 1c, only after the voltage of the stimulus pulse exceeds the inherent thresholds of the DUT, here identified as $V_{th+} = 1.45$ V and $V_{th-} = -1.65$ V, respectively. The inherent threshold voltage of $TiO_x$ devices in our case varies in the range of approximately $\pm 0.6$ V–2.5 V (see Supplementary Fig. 2).

**Hardware infrastructure.** The biasing protocol was implemented using custom made hardware developed in-house (Supplementary Fig. 4). It consists of a microcontroller-based printed circuit board (PCB)-mounted system[43] capable of addressing devices embedded in crossbar arrays of up to 1 kb in size ($32 \times 32$). The system has the capability of either testing packaged arrays or communicating to a multi-channel probe card for direct testing on the wafer. The hardware is supported by custom-made software that permits exhaustive, device-by-device testing of entire crossbar array or an array of individual devices in one, fully automated round of measurements. The biasing schemes applied for read and write operations are the $V_r$ (Fig. 3 in ref. 44) and $V_{r/2}$ (Fig. 10b in ref. 45) schemes, described in detail in their respective references. This helps in mitigating the sneak path effects.

**Mathematical model.** For the DUT, curve-fitting was carried out using standard curve-fitting tool in MATLAB. The data from the resistive state of the devices for negative (Fig. 1c) and positive (Fig. 1d) pulses were separately fitted to second-order exponential function, that is $f(\int V dt) = A e^{\beta \int V dt} + B e^{\gamma \int V dt}$, where $V$ is the fixed pulse voltage indicating non-volatile resistive states transitions. The data for the mathematical model is tabulated in Supplementary Information (Supplementary Table 1).

**CMOS MEA.** Neural activity from the portions of dissected mid-peripheral rabbit retinal ganglion cells was recorded using extended CMOS technology[35,36] (Supplementary Fig. 3). The surface of CMOS multi-transistor array comprising of $128 \times 128$ sensor sites is insulated by a thin, inert $TiO_2/ZrO_2$ layer. A thin metal layer beneath the oxide layer is connected to the gate of the field-effect-transistor via metallic pathway. The source drain current of the MOSFET in the silicon-based field effect transistor is modulated by the application of local voltage changes within the interfaced neural tissue above the recording sites. The CMOS MEA termed as front-end consists of the MEA itself, which operates at a 12.2 k frame per second sampling rate and outputs current time-series in blocks of $\sim 63$ k samples. The board-mounted TIAs convert the signal into voltage and boost it from the 0.1 mV–1 mV to the 10 mV–100 mV range. There was no modification on the front-end system and the pre-recorded blocks of dissected rabbit retinal ganglion cells placed atop the chip are measured.

**MEA neural recording signal-processing.** In the implementation of the neural activity sensor used for our experiments, an external front-end of the MEA-based CMOS system in ref. 31 was used (see Methods section, CMOS MEA and Supplementary Fig. 3). Each neural recording is 63 k samples in length and was fed to an in-house developed memristor characterization instrument[31,43]. The customized hardware handled the software-implemented linear gain and offset conditioning operations, electrically interfaced test memristors (Supplementary Fig. 4) and carried out the DUT resistive state assessment procedures (Fig. 2b and Supplementary Fig. 5). Neural signal voltage time-traces were fed into the target device in batches of 1,000 data-points. Resistive state was assessed at the beginning of each batch, then every 300 samples and at the end of the batch (standard scheme: assess initial resistive state and after application of the 300th, 600th, 900th and 1,000th data-points). Since the events are transduced as non-volatile resistive state transitions one can afford smaller sampling rates that benefits further time-resolution data-rate. Subsequently, changes in DUT resistive state ($\Delta R$) can be extracted from pairs of consecutive resistive state readings, while resistive state changes occurring between the last measurement of each batch and the first measurement of the next batch, that is with no interceding pulse biasing, (N) provide an estimate of measurement uncertainty thus generating the noise band. Thus, for a single neural recording we obtain 316 $\Delta R$ values, corresponding to 252 $\Delta R$ bins and 64 noise level sample which helps in determining the extracted thresholds ($V_{eth-}$) separating significant from insignificant resistive state switching activity (Supplementary Fig. 6). The range of extracted threshold voltages for $TiO_x$ family in our case is $-0.8$ V to $-1.8$ V (Supplementary Fig. 2). Importantly, noise band limits are set using the 6σ method that is, mean ($\mu$) ± three s.d. ($\sigma$) of noise level samples. Everything outside the noise band is considered as a significant resistive state change. Moreover, measuring the noise band helps in filtering out the insignificant resistive state changes caused due to weak amplitude neural signals. Furthermore, since the MIS system detects normalized changes in the resistive state rather than absolute values, the device variability is heavily compensated for such that MIS operation is routinely available.

**Data availability.** All data supporting this study are openly available from the University of Southampton repository at http://dx.doi.org/10.5258/SOTON/400411.

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

## Acknowledgements

We acknowledge the financial support of FP7 RAMP and EPSRC EP/K017829/1. Experimental procedures involving the use of animals were approved within the RAMP projects by Ethics Committee of the University of Padova and the Italian Ministry of Health (authorization.447/2015-PR). All the experiments were conducted in accordance with the approved guidelines.

## Author contributions

T.P. and S.V. conceived the experiments. A.K. fabricated the samples. I.G. and A.S. performed the electrical characterization of the samples and developed the control instrumentation and software. R.Z. developed the front-end recording platform. All authors contributed in the analysis of the results and in writing the manuscript.

## Additional information

**Competing financial interests:** The authors declare no competing financial interests.

