## [Peer Review File · Nature Communications]

Reviewers' comments:

Reviewer #1 (Remarks to the Author):

A. Summary of key results

-This paper uses memristor devices as "thresholded integrators" of voltage signals, and exploits this property to reduce the amount of data that must be transmitted from a neural recording chip to the external world. The idea is a clever one. Whenever the voltage from a neural spike exceeds a certain threshold, the resistance of the memristor changes, and holds (maintains) that change over time. Thus, if one periodically measures the memristors's resistance, one can determine how many spikes occurred between any two such measurements. Importantly, these measurements can be done at some rate of practical interest, much slower than the > 1 kHz sampling rates needed to detect each spike. Instead of sending the > 1 kHz waveform, one can then send a slower sequence of resistance measurements. In the paper they choose in one case to read out the memristor state at a frequency of about 1 Hz. Transmitting information slowly in this way could save greatly on data-transmission-related power expenses and hence on tissue heating and other key properties of implantable BMI devices.

-This is worth publishing, in my opinion, but the clarity of the exposition in the paper, particularly the figures, needs to be improved before publication.

B. Originality / Interest

-As far as I know, this is a novel idea, and the authors have gone a long way towards showing that it can be implemented in a real device.

-Data compression is important technologically from the perspective of brain machine interfaces, and thus this paper contributes a novel potential solution to a relevant problem.

-This study is novel, relevant and deserves to be published. Before it can be published, though, the paper needs to be made much clearer and better explicated. The figures are far too crowded, and are difficult to understand. The few most important elements need to be highlighted, and the rest removed or moved to the supplement.

C. Data and methodology

-Visually in the figures, I see some super-threshold spikes which do not lead to a visible resistance change. Why? This appears to be reflected in Figure 3c. Spikes slightly above or below the threshold seem to cause a response, whereas spikes right at the threshold do not. Would this limit the practical use of the device?

-Also, there appear to be some background fluctuations in the resistance. What causes them and how can they be limited? What effect do they have?

D. Appropriate treatment of uncertainties

- "All of our TiOx devices exhibited similar threshold voltages, even though the precise values among them varied." This needs to be made more specific. Varied by how much? What is "precise"?

E. Conclusions

- I am curious about why using memristors for this purpose is superior to other potential ways of constructing a "thresholded integrator" circuit on chip, for the purpose of data compression. Presumably their low power consumption is the main potential advantage? But on the negative side, see the point about spike-sorting below.

- One point that needs to be made, relevant to the practical use of this type of device, concerns the local field potential (LFP). Electrodes typically record both spikes (on the order of 100 μ V) and a slowly varying background potential (the LFP) which may be quite large compared to the spikes. This background needs to be filtered off by the front end.

- Another point is about spike sorting. Often, devices record the full >10 kHz waveform on an electrode because this full waveform allows distinguishing which neuron gave rise to a given spike, based on spike shape. This is no longer possible when spike detection is done via thresholding in this way. The authors should make note of this limitation.

- The challenges or downsides associated with using memristors need to be more clearly explained as well. They talk about this study as paving the way for more power, area and bandwidth efficient recording systems, but what are the key next steps?

- More should be said about the estimated fold-reduction in data transmission bandwidth that can realistically be achieved in this approach, e.g., in the multiplexed design of Figure 5. The section in the discussion on this is unclear. In that discussion, it seems that the state of the art CMOS recorder is actually winning on power dissipation over the memristor device (2.6 μ W vs. 30.5 μ W), which seems concerning, although I may be misunderstanding.

F. Suggested improvements

- The clarity of the writing and exposition needs to be significantly improved. There is far too much going on in the figures.

- There is too much emphasis on technical aspects of memristors before any clear overall visual communication of the "main idea", which to me is basically what I summarized in section A of this review, above. There should be a figure that summarizes conceptually this basic idea. Perhaps a schematic showing the *desired* behavior, similar to Figure 3a and 3b.

- And then the remaining figures should be stripped down considerably, or moved to the supplemental.

G. References

-No comments

H. Clarity

-In "A corollary of Chua's original definition of the memristor is that such elements are capable of changing their RS", "RS" needs to be defined in words.

-What is meant by the phrase "invasive", e.g., "invasive trains of programming pulses"?

Reviewer #2 (Remarks to the Author):

In their paper entitled "Towards a 'smart' brain-chip interface: real-time encoding and compression of neuronal spikes by metal-oxide memristive devices" the authors propose the use of memristive devices in the context of high-density brain chip interfaces (BCIs) with the memristors used for spike-counting purposes. Some experimental single device data and a scenario for high-density BCIs are presented. Whereas the latter case is clearly the vision the paper is aiming for ("smart" BCI), unfortunately this topic is still on a proposal level only and a few considerations made in this context should be scrutinized and reworked.

Overall the paper is well organized and well written, and the idea of the paper is definitely exciting and new. However, as stated above, there are some minor and some major weaknesses which should be re-considered by the authors before the paper is considered for publication in Nature Communications. A more detailed list is given below following the order of occurrence of the related issues.

Page 2, 2nd paragraph, last two lines:

Here it is stated that all devices exhibit "similar" threshold voltages after the respective treatment.

Please provide that information using quantitative data, i.e. magnitude of ensemble considered and 1-sigma or 3-sigma standard deviation.

Page 2, last paragraph, upper half:

Please explain "ad-hoc chosen amplitude".

Page 2, last but 6th line, "... input voltage integral (magnetic flux) ...":

Of course voltage integrated over time has the unit Vs, which is also the unit of magnetic flux, but - physically speaking - are the phenomena which apply here interpreted as magnetic flux-related in an (quantum?) electrodynamics-related sense?

If yes, please provide a reference, if no, please skip the parentheses and the term in it.

Page 3, line 8, + several further times, expression "MTA":

Indeed, that expression is used in the literature referred to, however, from a designer's standpoint it reads a little misleading (as all CMOS MEAs consist of arrays of transistors, e.g. also refs. 3, 4, ...).

It is thus recommended to replace this expression by the more general term "CMOS MEA".

Page 4, section "Towards array-level operation", Fig. 5 and related discussion:

To extend the scheme shown in Fig. 5 and applied to a MEA system with 224 (=16 x 14) recording sites to large MEAs, let us - for simplicity - consider a device with 100 x 100 sites and a full frame sampling rate of 10 kHz.

Multiplexing the entire analog data stream through one TIA + offset adder would require to operate these components at a BW of order $10 \text{ kHz} \times 100 \times 100 \times 10$ (last factor 10 to take into account settling times!) = 1 GHz. This is not realistic.

Thus a column-parallel or row-parallel (as in CMOS APS devices referred to in this section at the beginning or as in ref. 11) approach could be applied, translating into a required column- or row-related bandwidth of approx. 10 MHz (= $10 \text{ kHz} \times 100 \times 10$), a sampling rate per column or row of order 1 MHz, and write windows of the order of several 100 ns, which seems feasible following the argumentation of the paper later on (cf. Discussion).

Are these thoughts correct or is there something misunderstood?

If correct, the figure and the respective sections should take into account and represent these considerations, i.e. the figure should be changed / extended and the text should be modified and pick up these thoughts.

Page 5, last but 5th line, data compression with respect to spike amplitude:

Does that mean, information can be coded / sampled representing the magnitude of the amplitude or is only thresholding possible (i.e. single bit information)?

Page 6, top:

The energy / power discussion is "not fair", as it only considers the energy used to operate the memristors themselves but not the writing and reading circuitry. Given the relative high BW in the writing path (cf. discussion above), this (analog) path will burn an amount of power which may not yet be predictable at this point in time but which must surely not be ignored.

In conclusion, the energy / power issues of a memristive array of course may be mentioned in the manuscript but it must be clarified that the respective numbers do not speak for a system level implementation! Challenges of the related peripheral circuitry should at least be mentioned in a qualitative statement as well.

Page 6, 1st paragraph, 2nd half:

Here, the entire text suffers from a lack of clarity. Some related questions are given below:

- Does the output of the envisioned CMOS MEA use sigma-delta modulators, is that part of your scenario? Otherwise, other ADC topologies are attractive for BCI applications, e.g. SARs are frequently used.

- If a sigma-delta approach is applied and following the discussion above, a single

modulator alone will run into BW limitations, i.e. you would need multiple (column-wise or row-wise) modulators.

- What is meant by " 'single' voltage time-series sample"? The signal at the output of a sigma-delta modulator is usually referred to as "pulse-train", and the signal is of binary nature. In the case, a multi-bit modulator topology is used, of course that kind of signal can be forwarded as well. Please clarify.

- Realization of the MIS system atop the CMOS MEA: Obviously, this will run into spacing problems, as on top of the CMOS MEA circuitry usually the interfacing sites are arranged. Your scenario may apply under condition that multiple additional extra processes are applied after CMOS fabrication, i.e. MIS first followed by the interfacing technology. This, however, will translate into huge technological efforts and challenges.

Moreover, given that idea, if the MIS layer is arranged on top of the CMOS device with memristors in every recording site, there will be a problem to implement the MIS operating circuitry within the same site.

In conclusion, although this scenario sounds very attractive, there are also significant hurdles with no solution on the horizon so far, so that it is recommended not to include such ideas without mentioning the challenges or to briefly mention them but strongly emphasizing the open points.

Methods Summary, explanation of the CMOS MEA, middle:

Electronically speaking, it is not the sensing oxide layer which is connected to the gate of a sensing FET through a metallic pathway, but a thin metal layer ("electrode") underneath that sensing oxide. Please correct.

Fig. caption of Fig. 5 and related text:

You refer to a 14 x 16 CMOS MEA in standard CMOS technology, the references given here, however, are related to a device with several 1000 sites. Please clarify.

Fig. 1: The green text in the upper left figure is hardly legible which may originate from a pdf artefact. Please check and / or improve.

Figs. 2, 3, 4: The depicted neural signals show amplitudes in the V range and in part an offset in the several 100 mV range. Given the magnitude of real recorded signals, the signals shown here must be amplified by roughly 60 dB and maybe further processed. Please clarify (in the figure captions and / or in the manuscript).

Fig. 4: c), d), g), and h) are hardly legible. Recommendation: Increase the respective traces.

Reviewer #3 (Remarks to the Author):

In this work, the authors have demonstrated a very interesting and unique application of memristive devices to record synapses by means of integrating voltage pulses with amplitudes larger than a pre-established value (i.e., by adding an offset that can trip a recording event when the offset plus the stimulus sits above the memristor switching threshold voltage). The idea is very clever and interesting, and in fact the authors have already posted this work in the archives ([1], <https://arxiv.org/abs/1507.06832>).

I will describe why I believe this paper is suitable for publication:

1) Although the fact that memristors can "accumulate" a history of pulses by incrementally changing its resistance (see, for example, ref. [2]), as far as I know, this is the first proposal and implementation of the usage of memristors for recording synapse activity;
2) Prior CMOS interfaces to neural tissue [3] only addressed their corresponding noise behavior. The power spectrum density changes upon activity (or not) were used to describe that it is possible to extract patterns from neural tissue, however no classification was utilized, and furthermore, beyond this issue the method was computationally intensive. The authors basically by putting 1 and 2 together, demonstrated how one can minimize processing time, use memristor non-volatility in similar ways that long phosphor persistence was utilized in oscilloscopes back in the day, discriminate pulse heights [1] by the intrinsic exponential response of TiO₂ devices, and spatially locate this activity in regions that can be scaled down significantly. This summarizes the proposal and accomplishments of this work. Yet, I do have some issues that I see as important for a broader readership, myself included. The paper is not as organized as it can be. In order to help the authors, I will list what can be done to improve clarity (the fact that maybe some of what I will write could be already in the paper only means that it can be made better):

1) The overall system description is confuse and misleading. I am not sure what the experiment consisted of, whether the whole integrated system (retina+cleft+CMOS+memristor integrator sensor array+biasing&MUX network) was utilized, and how, or parts of it were put together separately and stitched together by software (in particular I was lead to think that way by the sentence: "Pre-recorded blocks of spontaneous rabbit retinal ganglion cell activity, 64kS in length, were fed to an in-house developed memristor characterization instrument"). When looking at the group's past reference (see for example ref.[4]), I see that there is a biasing/write/read system that can set the unselected rows and columns appropriately for cell readout (with a 10% error). In regards to that:

a. How was this used?

b. Is it of mandatory usage?

c. How can one run the system with all memristors cell recording synaptic data in parallel? By means of the MTA?

d. If the memristor integrator array is used for integrating simultaneously all synaptic activity (I don't think it is, I would like to see if it is possible), how one deals with multiplexing, half-selected voltages, write disturb from adjacent cells, and all problems that plague 1R memristor array architectures? I understand that in order to make this first and important step one needs a MUX and a DEMUX (Fig. 5) to answer the above question, but one possible goal could be getting rid of all that. Can the authors further elaborate on that?

e. Fig. 3 helps but does not answer all the questions above; fig 5 is a bit more clear, but I am still not sure whether the data flow from cells to memristor was carried out directly as fig. 5 indicates, or if there was an intermediate step to record the data, and then further

written by demultiplexing it into the memristor integrated sensor.

2) I believe that an overall picture of the system, including the culture cell is important. As mentioned above, the paper is not very clear in saying that fig. 5 is the end goal, and that an intermediate process is carried out (of pre-recording data, and writing data into the memristor array)

Clarifying the above points is mandatory prior to publication. Also points 1.a-d help understand that the idea proposed can be scaled up, and that is the key advantage in contrast to existing approaches.

[1] <https://arxiv.org/abs/1507.06832>

[2] Switching dynamics in titanium dioxide memristive devices, Pickett, M.D. et al, Journal of applied Physics, 106, 074508 (2009).

[3] Zeitler, R., Fromherz, P. & Zeck, G. Extracellular voltage noise probes the interface between retina and silicon chip. Appl. Phys. Lett. 99, 263702 (2011).

[4] Berdan, R. et al. A u-Controller-Based System for Interfacing Selectorless RRAM Crossbar Arrays. IEEE Trans. Electron Devices 62, 2190-2196 (2015).

Reviewer #1 (Remarks to the Author):

Summary of key results

-This paper uses memristor devices as "thresholded integrators" of voltage signals, and exploits this property to reduce the amount of data that must be transmitted from a neural recording chip to the external world. The idea is a clever one. Whenever the voltage from a neural spike exceeds a certain threshold, the resistance of the memristor changes, and holds (maintains) that change over time. Thus, if one periodically measures the memristors's resistance, one can determine how many spikes occurred between any two such measurements. Importantly, these measurements can be done at some rate of practical interest, much slower than the > 1 kHz sampling rates needed to detect each spike. Instead of sending the > 1 kHz waveform, one can then send a slower sequence of resistance measurements. In the paper they choose in one case to read out the memristor state at a frequency of about 1 Hz. Transmitting information slowly in this way could save greatly on data-transmission-related power expenses and hence on tissue heating and other key properties of implantable BMI devices.

-This is worth publishing, in my opinion, but the clarity of the exposition in the paper, particularly the figures, needs to be improved before publication.

We thank the referee for acknowledging the quality of our results by supporting that this is 'worth publishing.' We have taken the referee's comments on board, in particular revamping the content (figures and discussion) of this manuscript for improving clarity, as specifically addressed below.

B. Originality / Interest

-As far as I know, this is a novel idea, and the authors have gone a long way towards showing that it can be implemented in a real device.

-Data compression is important technologically from the perspective of brain machine interfaces, and thus this paper contributes a novel potential solution to a relevant problem.

-This study is novel, relevant and deserves to be published. Before it can be published, though, the paper needs to be made much clearer and better explicated. The figures are far too crowded, and are difficult to understand. The few most important elements need to be highlighted, and the rest removed or moved to the supplement.

We thank the referee for noting the key contributions of this work. We have now streamlined the presentation of this work, including Figures, for ensuring the widest possible accessibility of content without compromising on the quality of the presented data. More specifically:

- Figure 1 shows the device architecture and electrical characterisation of solid-state TiO_x ReRAM devices. All colours have now been selected for improving visibility (including in black and white printouts) and the highlighted band in Fig.1c was introduced to indicate the onset of resistive switching.
- Figure 2 has been considerably changed and now illustrates at first the Memristive Integrating sensor (MIS) concept and its operation as an integrating sensor. The MIS block diagram has now been simplified along with the corresponding caption to distinguish MIS from the externally employed front-end system for recording neural activity, also supported by Fig. S3. The block diagram for our benchmarking 'Template matching system' (initially Fig. 2c) has been moved to supplementary information (Fig.S7). The previously employed figure (initially Fig. 2 d) for describing our read out scheme is now replaced by a simplified version appearing

as Fig. 2b; while the original, more detailed, figure is maintained as supplementary material (Fig. S5) for completeness.

- Figure 3 displays the experimental results for benchmarking our memristor-based system against the state-of-the-art template matching system. The raster plots depicted in Fig.3 c,d,g,h are now magnified for clarity.
- Figure 4 is now heavily updated to include in a coherent manner data that overall showcases the robustness of our devices; addressing a common remark. Subfigures (c,d,g,h) now include calculated quantification parameters (true/false +ve) for each case as a performance metric, detailed in Table ST2 and Fig. S6 in supplementary information.
- Figure 5 demonstrates the extension of the MIS concept at array level.

Besides, the manuscript has been considerably re-written with the aim to articulate more clearly our contributions and their novelty. We maintained all introduced concepts in the main manuscript with all experimental procedures and related technical details now appearing under the Methods and/or supplementary sections. For clarity, the title of section 3 in the main manuscript on Page 3 has been changed to ‘MIS system performance.’ Moreover, a new section on ‘Device Characterisation’ has been added in Methods.

C. Data and methodology

-Visually in the figures, I see some super-threshold spikes which do not lead to a visible resistance change. Why? This appears to be reflected in Figure 3c. Spikes slightly above or below the threshold seem to cause a response, whereas spikes right at the threshold do not. Would this limit the practical use of the device?

-Also, there appear to be some background fluctuations in the resistance. What causes them and how can they be limited? What effect do they have?

We thank the reviewer for identifying these important points. Indeed, previously in Figure. 3c (now Figure 2 c,d) there was a discrepancy in the marked threshold voltage and the visible resistance changes that rightfully led to this confusion. This is now rectified in the amended manuscript; more specifically:

- The experiment described in Fig. 1b,c, where a train of positive/negative pulses is applied at increasing voltage amplitude gives an approximate estimation of our devices’ threshold voltage. This is now referred to as “inherent” threshold voltage of device, on Page 2 paragraph 1. This is now further described under Methods in ‘Device Characterisation’: “*The inherent threshold voltage of TiOx devices in our case varies in the range of approximately $\pm 0.6-2.5V$ (see supplementary Fig. S2)*”. This denotes the switching threshold of our prototypes and is a fundamental device property, as described in detail in Fig. S2. Knowing this is crucial for setting the gain and offset during the neural data pre-processing stage.
- In-operando the MIS system detects spikes by examining changes in resistive state of the core memristors and deciding on the state modulation significance. The latter is important for disentangling resistive changes arising in response to actual spiking events versus minor background fluctuations due to measurement/instrument noise and/or device state drift. In our approach, this is particularly addressed by defining an “effective threshold”, as illustrated in Fig. S6. Moreover this is now more explicitly defined in the main manuscript, on Page 3, paragraph 4, under ‘MIS system performance’ section: “*This is better illustrated in Fig. 2e, where the normalised resistive state changes between consecutive ‘reads’ is plotted as a function of the maximum voltage magnitude of interceding events. The grey horizontal band marks resistive state changes that have been discarded (see methods section). The remaining points are used to define the memristor’s ‘effective, operating threshold voltages’ ($V_{eth+/-}$), which partition the plot into three distinct areas: two of them correspond to significant*

resistive state modulation ($<V_{eth-}$ and $>V_{eth+}$) and the last one ($[V_{eth+}, V_{eth-}]$) containing resistive state changes that are indistinguishable from the estimated background noise. The range of effective threshold voltages for the TiOx prototypes employed throughout this study was -0.8 to $-1.8V$ (Fig. S2). Importantly, whilst the “inherent” threshold of the device performs a coarse filtering action, the “effective” threshold ultimately determines SNR.’ Additional changes in Fig. 2d are now in line with the extracted threshold voltages. For better clarity, inherent and effective threshold voltage is now differentiated as V_{th-}/V_{th+} and $V_{eth-/+}$ respectively. Moreover, we have now added under methods ‘MEA neural recording signal-processing’ section the following: “...for a single neural recording we obtain 316 (ΔR) values, corresponding to 252 ΔR bins and 64 noise level sample which helps in determining the ‘effective’ thresholds (V_{eth-}) separating significant from insignificant resistive state switching activity (Fig.S6). The range of extracted threshold voltages for TiOx family in our case is -0.8 to $-1.8V$ (Fig.S2). Importantly, noise band limits are set using the ‘ 6σ ’ method i.e. mean (μ) \pm three standard deviations (σ) of noise level samples. Everything outside the noise band is considered as a significant resistive state change. Moreover, measuring the ‘noise band’ helps in filtering out the insignificant resistive state changes caused due to weak amplitude neural signals. Furthermore, since the MIS system detects normalized changes in the resistive state rather than absolute values, the device variability is heavily compensated for such that MIS operation is routinely available.”

- The impact of resistive state fluctuations on the MIS performance, as denoted by the referee, can be further mitigated by optimising the offset value as indicated on Page 4, paragraph 1, exemplified via an example by optimisation of offset: “However, the presence of a few events in opposite polarity that exceed V_{eth+} , cause occasional resistive state drops. A clear example indicated by ‘ φ ’ in Fig. 3g,h can be observed at approximately 1.4s in Figs. 3a,b and Figs. 3e,f (expanded view) where the resistive state reduces from $\sim 4.5k\Omega$ to $\sim 4k\Omega$. However, optimising the value of V_{off} provides additional flexibility for compensating for this effect.”
- The utilised values of effective threshold throughout this work are now included under the methods section ‘MEA and recording and signal-processing.’
- Sub-scripts in Figs. 2, 3, 4 and the respective captions have also been changed to V_{eth-} .
- The introduction of appropriate gain and offset settings provide greater flexibility in optimising the MIS performance and alleviating the challenges imposed by device-to-device or cycle-to-cycle variability. This is now better captured via Fig. 4. The referee’s remark has further instigated additional studies on further limiting factors of our approach. Besides those denoted and addressed previously, we have further identified the programming saturation of our device within a certain resistive state band to be a potential issue. This is now addressed in the main manuscript under the Discussion section, paragraph 3: “Finally, the key focus for driving this work forward in the future is to improve on the detection accuracy rates. A plausible performance-limiting factor is the programming saturation of memristive devices. This can however be counteracted by optimising the main operational parameters i.e. gain and offset settings (Fig.S12) or via employing memory state resets, as depicted in Fig. S13.”

D. Appropriate treatment of uncertainties

-“All of our TiOx devices exhibited similar threshold voltages, even though the precise values among them varied.” This needs to be made more specific. Varied by how much? What is “precise”?

We apologise to the referee for our abstract description. We have now provided additional experimental data relating to both the “inherent” and “effective” thresholds throughout this work, as depicted in Fig. S2. Specifically, it can be observed that the range for the inherent threshold varied between ± 0.6 - $2.5V$ as determined by data stemming from 10 devices while the range of effective

threshold varied between – 0.8 to -1.8V as determined by sampling data from 34 devices. We further note that in these experiments all neural recordings exhibited negative polarity spikes, thus only negative effective threshold data is available.

E. Conclusions

-I am curious about why using memristors for this purpose is superior to other potential ways of constructing a "thresholded integrator" circuit on chip, for the purpose of data compression. Presumably their low power consumption is the main potential advantage?

As pointed by the referee, the main advantage of our proposed MIS concept lies in its low power consumption, when compared to the present state-of-art (see *Paraskevopoulou, S. E. & Constandinou, T. G. A sub-1 μ W neural spike-peak detection and spike-count rate encoding circuit. in 2011 IEEE Biomed. Circuits Syst. Conf. 29–32 (IEEE, 2011). doi:10.1109/BioCAS.2011.6107719*). Additional benefits however are envisioned by the technology's scalability prospects and back-end-of-line integrability. We concur with the referee that is essential to offer a comparison of our approach to the state-of-art, now added as last paragraph on page 5: *"The required power budget of our approach can be optimised by mapping the highest amplitude neural recording samples onto ~5V pulses at <100ns due to the known voltage-time trade-off⁴¹ (see supporting Fig. S11). Under the realistic assumption of operating devices at resistive states of 100k Ω , every 1000 samples (one data batch, -as per the standard schematic-) we would spend a maximum of 250 μ W*100ns = 25pJ for biasing a device with neural recording data. Simultaneously, the read-out operation would cost 0.8 μ W*100 μ s = 80pJ (based on 0.4V read out voltage), rendering an average power dissipation of ~300nW per channel. Clearly, the memristor read-out and biasing circuitry will require an additional power. Nevertheless, estimated figures are already significantly less than the present state-of-art continuous time spike-detectors¹⁵ (~720nW-digital input). Most importantly, the proposed technology is demonstrated here at a proof of concept-level via large prototype devices and clearly the presented power/density considerations are not a reflection of the technology's full potential. We can expect that substantial improvements in power consumption can be achieved by further downscaling and/or operating memristors at even higher resistive state ranges, for e.g. operating the device in 1M Ω region can further reduce the power dissipation remarkably by two orders of magnitude. In addition, the bandwidth required to assess the resistive state should be rather low as compared to the raw input data-rate. Additional power efficiency gains can be expected by integrating the MIS elements atop state-of-the-art CMOS thus minimising parasitic capacitances."*

One point that needs to be made, relevant to the practical use of this type of device, concerns the local field potential (LFP). Electrodes typically record both spikes (on the order of 100 μ V) and a slowly varying background potential (the LFP) which may be quite large compared to the spikes. This background needs to be filtered off by the front end.

We agree with the reviewer and wish to further note that there was no modification in the front-end system; the local field potentials are indeed filtered off by using a high-pass filter. For the sake of clarity, the block diagram in Figure 2a has been edited accordingly to show that an external front-end system is used throughout this work that was interfaced to the MIS platform. This has now been noted on Page 2 'Neural spiking integration with metal-oxide memristors,' paragraph 1 describing the MEA as 'external'. Moreover, in Methods section now (CMOS MEA), it has been added that, *'there was no modification on the front-end system and the pre-recorded blocks of dissected rabbit retinal ganglion cells placed atop the chip are measured.'* Additionally, Fig S3 further drives this point clearer. Clearly the long-term vision of this work is towards the monolithic integration of the MIS system along with the neural recording platforms for performing in-situ spike-detection.

-Another point is about spike sorting. Often, devices record the full >10 kHz waveform on an electrode because this full waveform allows distinguishing which neuron gave rise to a given

spike, based on spike shape. This is no longer possible when spike detection is done via thresholding in this way. The authors should make note of this limitation.

The reviewer makes a valid point, however it must be noted that memristors are more than simple integrators yielding simple yes/no outputs. Indeed, in this study we present a novel approach for detecting spikes and approximate timing of the events occurrence that can be extrapolated from monitoring the transient response of memristive states of individual devices. We concur with the referee that spike-sorting is also important in brain-chip interfaces and although our approach of thresholding may appear as limiting to spike sorting that is not necessarily true. The fact that changes in resistive state are analogous to the signal (events) envelope allow us to preserve some of the signal richness and one can argue that sorting of spikes can be extrapolated via the magnitude of $\Delta R/R_0$ plots that we observe per event. This, however, requires an unambiguous knowledge of device characteristics that are identical for our devices in an array in order to be able to decompress this information reliably and across different recording devices. As such we can expect that certain amount of information on event amplitude is preserved but substantial further work is required towards reliably capturing this. Our group is indeed actively pursuing research in this auxiliary niche area, which is however at a preliminary stage. We have now added a few remarks for showcasing the future work in this niche area that can be found in the Discussion section on Page 6, paragraph 2, which says, *‘Another possible line of investigation towards MIS-based spike-sorting lies in determining how much information on spike amplitude/duration can be extracted from the magnitude of resistive state changes’* and in summary section which says, *“Moreover, as the memristor resistive state changes are linked to amplitude and polarity of the input waveform (signal envelope) this information is preserved in the magnitude of resistive state modulation.”*

-The challenges or downsides associated with using memristors need to be more clearly explained as well. They talk about this study as paving the way for more power, area and bandwidth efficient recording systems, but what are the key next steps?

We thank the referee for this remark, which we hope we have now addressed in our discussions section of the manuscript on Page 6; specifically:

(a) At present the foremost challenge is to improve the detection accuracy of the system, as now mentioned in the discussion section. We have now introduced new ways to improve this by optimising the used gain and offset settings (see Fig. S12) and employing frequent resets in the initial state of the device upon saturation (see Fig. S13).

(b) Operating the devices in higher resistive state regime to obtain increased power efficiency, as discussed in section E above and on Page 5 last paragraph.

(c) Scaling of the devices towards improving area-efficiency (Page 6, paragraph 1).

(d) Finally, working towards integration of the MIS system with the state-of-art neural recording platforms as now mentioned on page 4, last paragraph: *“We foresee that, a practical implementation of a monolithically integrated system will involve addressing the challenges associated with the integration of a MIS array with CMOS-based front-end circuitry, while the required MIS control can be accommodated as peripheral circuitry with sneak-path issues existing in dense RRAM crossbar configuration mitigated via selector topologies⁴⁰.”*

-More should be said about the estimated fold-reduction in data transmission bandwidth that can realistically be achieved in this approach, e.g., in the multiplexed design of Figure 5. The section in the discussion on this is unclear. In that discussion, it seems that the state of the art CMOS recorder is actually winning on power dissipation over the memristor device (2.6 uW vs. 30.5 uW), which seems concerning, although I may be misunderstanding.

The referee is correct in identifying that more needs to be said with regards to the BW/power benefits of our approach. The power benefits were argued in our previous responses, while we feel that the referee's remark related to bandwidth is now addressed by adding on page 4 the following: *"The concept introduced in Fig. 2a, when directly interfaced with front-end-circuitry, can be exploited for advancing the present state-of-art in high density neural recording platforms³⁸. The presented concept is amenable for scaling to a multi-channel array level, as illustrated in Fig. 5a, for capturing the activity of neural networks in real-time. We envisage an overall system architecture very similar to standard Active Pixel Sensor (APS) CMOS imagers³⁹. In this hybrid system, data from each of the 'N' pixels in the array arrives as an analogue current from the MEA and is multiplexed onto one of the 'M' on-chip trans-impedance amplifier (TIA) blocks, which are followed by on-chip offset stages. Thus, a small number of both gain and offset stages are time-shared by every pixel in the array. The conditioned recording data points are then de-multiplexed to a memristor bank, that can be integrated into the back-end of the chip, in good proximity to the MEA recording sites. MIS output is then generated by sequentially measuring the resistive states of each memristor in the bank. The low frequency at which memristor read-outs are generated (e.g., 200x lower data rate vis-à-vis input stream arriving from the MEA if 'standard scheme is used as described in methods section) allows the MIS system to carry out all measurements through a single, time-shared TIA feeding into a single Analogue-to-Digital Converter (ADC). The digitised results are then sent off-chip."*

F. Suggested improvements

-The clarity of the writing and exposition needs to be significantly improved. There is far too much going on in the figures.

-There is too much emphasis on technical aspects of memristors before any clear overall visual communication of the "main idea", which to me is basically what I summarized in section A of this review, above. There should be a figure that summarizes conceptually this basic idea. Perhaps a schematic showing the *desired* behavior, similar to Figure 3a and 3b.

-And then the remaining figures should be stripped down considerably, or moved to the supplemental.

We appreciate the referee's concerns on aiding comprehension of the presented concept for which we have now implemented significant changes in the figures and the presentation flow. Figure 2, 3 and 4 have been modified considerably for communicating the MIS concept clearly. The specific changes are mentioned in our previous response under the 'Originality and Interest' section.

Moreover, the section on Page 2, 'Memristors as Event integrators', has been considerably edited in order to emphasise on the novelty of our approach whereas the technical details on our experimental methodology is sustained within the Methods and Supplementary sections.

G. References

-No comments

H. Clarity

-In "A corollary of Chua's original definition of the memristor is that such elements are capable of changing their RS", "RS" needs to be defined in words.

This has now been eliminated across the entire manuscript.

-What is meant by the phrase "invasive", e.g., "invasive trains of programming pulses"?

“Invasive” indicated sufficient strength for eliciting a change in the resistive state of the device. This has now been modified to ‘input programming pulses’ and correspondingly “non-invasive read operation” has been simplified to “read pulse”.

Reviewer #2 (Remarks to the Author):

In their paper entitled "Towards a 'smart' brain-chip interface: real-time encoding and compression of neuronal spikes by metal-oxide memristive devices" the authors propose the use of memristive devices in the context of high-density brain chip interfaces (BCIs) with the memristors used for spike-counting purposes. Some experimental single device data and a scenario for high-density BCIs are presented. Whereas the latter case is clearly the vision the paper is aiming for ("smart" BCI), unfortunately this topic is still on a proposal level only and a few considerations made in this context should be scrutinized and reworked. Overall the paper is well organized and well written, and the idea of the paper is definitely exciting and new. However, as stated above, there are some minor and some major weaknesses which should be re-considered by the authors before the paper is considered for publication in Nature Communications. A more detailed list is given below following the order of occurrence of the related issues.

We thank the reviewer for acknowledging the novelty of our work. As with any “exciting and new” technological concept there is always room for improvement towards maturing it to the point that it can be exploited in practical applications; in our case “in high density BCIs”. This round of reviewing has truly enabled us to work towards further improving this novel concept and yet we acknowledge that there is much more to be done for gaining the full benefits of this disruptive approach. Here, we are presenting this concept for the very first time and as such we appreciate that there will be other numerous pathways that this can be exploited.

Page 2, 2nd paragraph, last two lines:

Here it is stated that all devices exhibit "similar" threshold voltages after the respective treatment. Please provide that information using quantitative data, i.e. magnitude of ensemble considered and 1-sigma or 3-sigma standard deviation.

Indeed this is useful information that is now added in response to both referee 1 and 2 comments; please see previous response above. We have further clarified the distinction between “inherent” and “effective” threshold. Beyond our previous response please see the amended caption of Fig. S2 (c) where we state: “Range of extracted threshold voltage of the TiOx devices used (as described in section ‘Neural spiking integration with metal-oxide memristors’ in the main manuscript and Fig. S6). The histogram is plotted for a sample of 34 devices. Range of the extracted threshold voltages used for the experiment is -0.8V to -1.8V. The mean (μ) of the sampled data is equal to -1.4V. Assuming a Gaussian distribution, 6σ (mean plus three times the standard deviation in either directions), 4σ and 2σ is equal to (-0.2V, -2.5V), (-0.6,-2.2V) and (-1V,-1.8V) respectively”.

Page 2, last paragraph, upper half: Please explain "ad-hoc chosen amplitude".

“Ad-hoc chosen amplitude” refers to the programming pulse amplitude chosen on a case-by-case basis. Following the referee’s suggestion, this has been amended to ‘suitable amplitude to induce a change in the resistive state’ on Page 2, paragraph 1, line 12.

Page 2, last but 6th line, "... input voltage integral (magnetic flux) ...": Of course voltage integrated over time has the unit Vs, which is also the unit of magnetic flux, but - physically speaking - are the phenomena which apply here interpreted as magnetic flux-related in an (quantum?) electro-dynamics-related sense?

If yes, please provide a reference, if no, please skip the parentheses and the term in it.

As suggested by the referee, we have now omitted any mentioning to “(magnetic flux)” and used the following, less opinionated statement on Page 1, ‘Memristors as Event Integrators’, paragraph 1: “*As originally proposed by Chua, memristors’ are capable of changing their resistive state as a function of the integral of their input voltage, a phenomenon known as “resistive switching.”*”

Page 3, line 8, + several further times, expression "MTA": Indeed, that expression is used in the literature referred to, however, from a designer's standpoint it reads a little misleading (as all CMOS MEAs consist of arrays of transistors, e.g. also refs. 3, 4, ...).

It is thus recommended to replace this expression by the more general term "CMOS MEA".

We concur with the reviewer and this expression has now been changed to CMOS MEA across the entire manuscript.

Page 4, section "Towards array-level operation", Fig. 5 and related discussion: To extend the scheme shown in Fig. 5 and applied to a MEA system with 224 (=16 x 14) recording sites to large MEAs, let us - for simplicity - consider a device with 100 x 100 sites and a full frame sampling rate of 10 kHz. Multiplexing the entire analog data stream through one TIA + offset adder would require to operate these components at a BW of order 10 kHz x 100 x 100 x 10 (last factor 10 to take into account settling times!) = 1 GHz. This is not realistic. Thus a column-parallel or row-parallel (as in CMOS APS devices referred to in this section at the beginning or as in ref. 11) approach could be applied, translating into a required column- or row-related bandwidth of approx. 10 MHz (= 10 kHz x 100 x 10), a sampling rate per column or row of order 1 MHz, and write windows of the order of several 100 ns, which seems feasible following the argumentation of the paper later on (cf. Discussion).

Are these thoughts correct or is there something misunderstood? If correct, the figure and the respective sections should take into account and represent these considerations, i.e. the figure should be changed / extended and the text should be modified and pick up these thoughts.

We greatly appreciate the referee’s insight that we have now taken into account by mentioning in page 4: “*The concept introduced in Fig. 2a, when directly interfaced with front-end-circuitry, can be exploited for advancing the present state-of-art in high density neural recording platforms³⁸. The presented concept is amenable for scaling to a multi-channel array level, as illustrated in Fig. 5a, for capturing the activity of neural networks in real-time. We envisage an overall system architecture very similar to standard Active Pixel Sensor (APS) CMOS imagers³⁹. In this hybrid system, data from each of the ‘N’ pixels in the array arrives as an analogue current from the MEA and is multiplexed onto one of the ‘M’ on-chip trans-impedance amplifier (TIA) blocks, which are followed by on-chip offset stages. Thus, a small number of both gain and offset stages are time-shared by every pixel in the array. The conditioned recording data points are then de-multiplexed to a memristor bank, that can be integrated into the back-end of the chip, in good proximity to the MEA recording sites. MIS output is then generated by sequentially measuring the resistive states of each memristor in the bank. The low frequency at which memristor read-outs are generated (e.g., 200x lower data rate vis-à-vis input stream arriving from the MEA if ‘standard scheme is used as described in methods section) allows the MIS system to carry out all measurements through a single, time-shared TIA feeding into a single Analogue-to-Digital Converter (ADC). The digitised results are then sent off-chip. We foresee that, a practical implementation of a monolithically integrated system will involve addressing the challenges associated with the integration of a MIS array with CMOS-based front-end circuitry, while the required MIS control can be accommodated as peripheral circuitry with sneak-path issues existing in dense RRAM crossbar configuration mitigated via selector topologies⁴⁰.*”

Page 5, last but 5th line, data compression with respect to spike amplitude: Does that mean, information can be coded / sampled representing the magnitude of the amplitude or is only thresholding possible (i.e. single bit information)?

Please see our response to referee 1 regarding the “spike-sorting capability” comment (last comment on Page 4). In principle information on magnitude to some extent should be preserved. Memristors detect spikes on the basis of resistive state transitions sensitive to both the polarity and magnitude of the event as mentioned on Page 5, paragraph 2: “*We note that whilst the system in Fig. S8 outputs a spike count that is insensitive to the amplitude of the detected spikes, the proposed MIS array results into a ‘ratiometric change in resistive state’ that is strongly correlated to the ‘strength’ of the individual spiking events. This allows us to preserve information on both event amplitude and polarity, which in principle improves the data compression rate.*” This could be relevant to in-vivo experiments where the information is indicative of the distance between the recording electrode and the spiking neuron. This has also been summarised on Page 6: “*Moreover, as the memristor resistive state changes are linked to amplitude and polarity of the input waveform (signal envelope) this information is preserved in the magnitude of resistive state modulation.*”

Page 6, top: The energy / power discussion is "not fair", as it only considers the energy used to operate the memristors themselves but not the writing and reading circuitry. Given the relative high BW in the writing path (cf. discussion above), this (analog) path will burn an amount of power which may not yet be predictable at this point in time but which must surely not be ignored. In conclusion, the energy / power issues of a memristive array of course may be mentioned in the manuscript but it must be clarified that the respective numbers do not speak for a system level implementation! Challenges of the related peripheral circuitry should at least be mentioned in a qualitative statement as well.

We concur with the referee’s comment. This point related to the power dissipation by read and write circuitry has now been added to the modified discussion section on power calculations on Page 5, last paragraph. Firstly, we now compare the power-dissipation for memristors with real-time continuous state-of-art spike-detectors instead of the system level comparison, which we found out, is a fairer comparison at this stage. Secondly, for power dissipation by read and write circuitry, this specific line has been added: “(1) *Clearly, the memristor read-out and biasing circuitry will require an additional power. Nevertheless, estimated figures are already significantly less than the present state-of-art continuous time spike-detectors¹⁵ (~720nW-digital input).* (2) *In addition, the bandwidth required to assess the resistive state should be rather low as compared to the raw input data-rate. Additional power efficiency gains can be expected by integrating the MIS elements atop state-of-art CMOS thus minimising parasitic capacitances.*”

Moreover, the challenges related to the required peripheral circuitry are now mentioned on Page 4, last paragraph, last line in ‘Towards Array-level MIS operation’ which states, “*We foresee that, a practical implementation of a monolithically integrated system will involve addressing the challenges associated with the integration of a MIS array with CMOS-based front-end circuitry, while the required MIS control can be accommodated as peripheral circuitry with sneak-path issues existing in dense RRAM crossbar configuration mitigated via selector topologies⁴⁰.*”

Page 6, 1st paragraph, 2nd half: Here, the entire text suffers from a lack of clarity. Some related questions are given below:

- Does the output of the envisioned CMOS MEA use sigma-delta modulators, is that part of your scenario? Otherwise, other ADC topologies are attractive for BCI applications, e.g. SARs are frequently used.

We thank the referee for giving us the opportunity to clarify this confusing point. In the original manuscript, we have referred to a traditional CMOS based spike detection system purely for

comparison purposes to our MIS approach. This has now been fully replaced with a more suitable example (see page 5, last paragraph). Finally, we note that the CMOS MEA employed throughout this work for recording the neural activity does not include any sigma-delta modulators. Thus, any previous mentioning of sigma-delta modulators, in the earlier manuscript related to the old example used for benchmarking power consumption of the MIS platform, is now amended.

- If a sigma-delta approach is applied and following the discussion above, a single modulator alone will run into BW limitations, i.e. you would need multiple (column-wise or row-wise) modulators.

We agree with the reviewer and indeed this is why we do not use sigma-delta modulators in our approach. We have now further indicated that a CMOS APS imager-like system may be more applicable to this task, see page 4, 'Towards array-level MIS operation', first paragraph.

- What is meant by "'single' voltage time-series sample"? The signal at the output of a sigma-delta modulator is usually referred to as "pulse-train", and the signal is of binary nature. In the case, a multi-bit modulator topology is used, of course that kind of signal can be forwarded as well. Please clarify.

We apologise for this oversight that is due to our previous choice of benchmarking system (as explained previously). Replacing the comparison now rectifies this issue and thus any mentioning of "single voltage time-series sample" is naturally omitted. All remaining mentioning on "voltage time-series" refer to a stream of analogue values.

- Realization of the MIS system atop the CMOS MEA: Obviously, this will run into spacing problems, as on top of the CMOS MEA circuitry usually the interfacing sites are arranged. Your scenario may apply under condition that multiple additional extra processes are applied after CMOS fabrication, i.e. MIS first followed by the interfacing technology. This, however, will translate into huge technological efforts and challenges. Moreover, given that idea, if the MIS layer is arranged on top of the CMOS device with memristors in every recording site, there will be a problem to implement the MIS operating circuitry within the same site. In conclusion, although this scenario sounds very attractive, there are also significant hurdles with no solution on the horizon so far, so that it is recommended not to include such ideas without mentioning the challenges or to briefly mention them but strongly emphasizing the open points.

We concur with the reviewer that interfacing of MIS system monolithically with a CMOS MEA platform requires addressing a number of challenges. Hence, we have taken the reviewer's advice and introduced the concept briefly on Page 4, paragraph 3 along with the challenges associated with it. This line has been specifically added to the main text, "*We foresee that, a practical implementation of a monolithically integrated system will involve addressing the challenges associated with the integration of a MIS array with CMOS-based front-end circuitry, while the required MIS control can be accommodated as peripheral circuitry with sneak-path issues existing in dense RRAM crossbar configuration mitigated via selector topologies⁴⁰.*" We further note that taking into account an APS-like architecture, the MIS processing need not be co-located with the sensing sites, i.e. occupying a dedicated area on the chip and thus relax integration challenges.

Methods Summary, explanation of the CMOS MEA, middle: Electronically speaking, it is not the sensing oxide layer which is connected to the gate of a sensing FET through a metallic pathway, but a thin metal layer ("electrode") underneath that sensing oxide. Please correct.

We thank the referee for bringing this oversight to our attention. This is now corrected on Page 7, Methods section (CMOS MEA): "*A thin metal layer beneath the oxide layer is connected to the gate of the field-effect-transistor via metallic pathway.*"

Fig. caption of Fig. 5 and related text: You refer to a 14 x 16 CMOS MEA in standard CMOS technology, the references given here, however, are related to a device with several 1000 sites. Please clarify.

We apologise for this confusion. Although, as the referee points, the employed CMOS MEA is capable of recording from an array of 128x128 sites, for the sake of demonstrating the MIS concept at the array level, we have restricted our study into a 14x16 sub-array, as mentioned Page 5, paragraph 1: *“In this work, this concept was validated via a hybrid approach that is capable of processing 224 distinct recording traces stemming from a 16x14 pixel subset of the previously employed MEA system^{2,34}, atop which retinal cells were residing. The sub-array*”

Fig. 1: The green text in the upper left figure is hardly legible which may originate from a pdf artefact. Please check and / or improve.

Amended accordingly.

Figs. 2, 3, 4: The depicted neural signals show amplitudes in the V range and in part an offset in the several 100 mV range. Given the magnitude of real recorded signals, the signals shown here must be amplified by roughly 60 dB and maybe further processed. Please clarify (in the figure captions and / or in the manuscript).

The acquired, pre-amplified neural signals from the front-end system are in the order of 10-100 mV range. This specific detail is added on Page 2, in ‘*Neural spiking integration with metal-oxide memristors*’: *“The MEA employed records the raw bio-signals, which lie in the 0.1-1 mV range, and then uses its own in-built amplifiers to boost them to the 10-100 mV range.”*

We particularly note that on top of the amplified/offset imposed by the CMOS MEA we have employed in software additional gain and offset as required, now clarified in the new text provided in page 2, last paragraph: *“The MEA employed records the raw bio-signals, which lie in the 0.1-1 mV range, and then uses its own in-built amplifiers to boost them to the 10-100 mV range. The resulting, boosted recordings are then stored off-line as voltage-time series. For this work, we have used these stored recordings as inputs to our MIS platform in isolation from the front-end, i.e. the front-end has not been connected to the MIS platform in real-time (Fig. S3). The processing of neural signals through MIS platform begins when the stored voltage-time series are subjected to amplification and offset in software on the PC that runs the platform (Fig. 2a - box ‘i’ and Methods section). This set-up offers the option of adjusting the MIS detection threshold and consequently allowing the integration of significant spiking events with a pre-determined SNR. For example in Fig. 2c, the offset and scaling parameters were chosen such that only the most significant events (i.e., largest amplitude extracellular spikes) would exceed the threshold. The resulting, pre-conditioned waveform is then transmitted from the PC to the memristor testing and operation instrument (see Methods, ‘Hardware Infrastructure’), which physically implements the MIS system. The instrument, in turn, plays back the waveform to a target memristive device.”* Moreover, specific gain and offset values are now provided in all relevant figures’ captions.

Fig. 4: c), d), g), and h) are hardly legible. Recommendation: Increase the respective traces.

Amended as suggested.

Reviewer #3 (Remarks to the Author):

In this work, the authors have demonstrated a very interesting and unique application of memristive devices to record synapses by means of integrating voltage pulses with amplitudes larger than a pre-established value (i.e., by adding an offset that can trip a recording event

when the offset plus the stimulus sits above the memristor switching threshold voltage). The idea is very clever and interesting, and in fact the authors have already posted this work in the archives ([1], <https://arxiv.org/abs/1507.06832>).

I will describe why I believe this paper is suitable for publication:

1) Although the fact that memristors can "accumulate" a history of pulses by incrementally changing its resistance (see, for example, ref. [2]), as far as I know, this is the first proposal and implementation of the usage of memristors for recording synapse activity;

2) Prior CMOS interfaces to neural tissue [3] only addressed their corresponding noise behaviour. The power spectrum density changes upon activity (or not) were used to describe that it is possible to extract patterns from neural tissue, however no classification was utilized, and furthermore, beyond this issue the method was computationally intensive.

The authors basically by putting 1 and 2 together, demonstrated how one can minimize processing time, use memristor non-volatility in similar ways that long phosphor persistence was utilized in oscilloscopes back in the day, discriminate pulse heights [1] by the intrinsic exponential response of TiO₂ devices, and spatially locate this activity in regions that can be scaled down significantly. This summarizes the proposal and accomplishments of this work.

Yet, I do have some issues that I see as important for a broader readership, myself included. The paper is not as organized as it can be. In order to help the authors, I will list what can be done to improve clarity (the fact that maybe some of what I will write could be already in the paper only means that it can be made better):

We thank the referee for acknowledging the quality of our results by supporting that this is 'suitable for publishing.' We have taken the referee's comments on board and have specifically addressed any concerns below.

1) The overall system description is confuse and misleading. I am not sure what the experiment consisted of, whether the whole integrated system (retina + cleft + CMOS + memristor integrator sensor array + biasing & MUX network) was utilized, and how, or parts of it were put together separately and stitched together by software (in particular I was lead to think that way by the sentence: "Pre-recorded blocks of spontaneous rabbit retinal ganglion cell activity, 64kS in length, were fed to an in-house developed memristor characterization instrument"). When looking at the group's past reference (see for example ref.[4]), I see that there is a biasing/write/read system that can set the unselected rows and columns appropriately for cell readout (with a 10% error). In regards to that:

a. How was this used?

We apologise to the reviewer for the lack of clarity. We hope the major re-writing of the manuscript has addressed this concern. More specifically, the employed front-end system (retina+cleft+CMOS) is now explicitly declared as external to the MIS system; mentioned both in the main manuscript and supplementary information as noted below:

1. Figure 2a (initially Fig.2b), the block diagram illustrating the MIS concept and operation has been edited. It has now been clearly mentioned that the retina+cleft+CMOS front-end was 'external' to the proposed system on Page 2, 'Neural Spiking integration with metal-oxide memristors', paragraph 2: "*We validated experimentally the MIS system implementation on spiking activity of retinal ganglion cells. At First, the activity of dissected retinal cells was pre-recorded by an external Multi Electrode Array (MEA) front-end system^{2,33,34,35,36} (see Methods, 'CMOS MEA').*" Moreover, throughout the manuscript we address "CMOS MEA" as "external."

2. The Fig. 2 caption has now been changed to: “*MIS concept and operation. (a) ... (i.e., a CMOS MEA system in our experimental implementation) located externally to the MIS platform records extracellular neuronal signals....*”
3. Fig. S3 has been added for better clarifying that the front-end system used was external to the proposed system.
4. The description of “MEA neural recording signal-processing” in the Methods section on Page 7 has been modified. It has now been specifically mentioned that, “***There was no modification on the front-end system and the pre-recorded blocks of dissected rabbit retina....***”

Moreover, the pre-amplified neural recordings obtained from the front-end (as shown in Fig. S3) were processed by giving a gain and offset based on the threshold of the devices. On pre-processing, these neural-recordings are fed to the devices using the characterisation board, shown in Fig. S4. This has now also been supported by additional figures in Supplementary material (Fig. S3,5,6), while the methodology description is amended accordingly, namely:

- The section on Page 2, ‘Neural Spiking integration with metal-oxide memristors’ has been completely re-written with the objective of clarity. Paragraph 1 describes the concept whilst paragraph 2 deals with the specific implementation of using devices as integrating sensors.
- Figure 2, has been completely edited to illustrate the concept and signal-processing methodology used for MIS operation. Figure 2 caption has been edited to: “*MIS concept and operation. (a) Block diagram of the signal processing in the proposed spike-detection system. An external frontend (i.e., a CMOS MEA system in our experimental implementation) located externally to the MIS platform records extracellular neuronal signals and amplifies them. The pre-amplified, acquired neural recordings are then fed into our instrument, suitably gain-boosted (G) and offset (V_{off}) to render them compatible with the memristors’ voltage operating regimes (i). The conditioned waveform is fed into a memristor and its resistive state is then periodically assessed (ii). Changes in resistive state caused by spiking events are extracted offline (iii). (b) Conceptual read-out scheme for evaluating the time evolution of the resistive state of test devices subjected to input stimulation for one batch. The resistive state (red line) is assessed at the beginning of each neural recording batch (blue trace), then every chosen number of samples (i.e., 200 in reported example) termed as ‘bin’ (B) and finally at the end of each batch (assessment points marked by crosses). Changes in test device resistive state (ΔR) are extracted from consecutive resistive state assessments. Resistive state changes occurring between the last measurement of each batch and the first measurement of the next batch, i.e. with no interceding pulse biasing, (N) are considered to result from measurement uncertainty and can be used to determine the noise band. (c), (d) Shows an arbitrary input waveform consisting of four concatenated copies of the same retinal cell recording (original copy with negative spikes), and artificially inverted to produce spike trains with alternating polarities. This waveform was employed to validate the concept of memristive integrating sensors, the response of which is shown in (d). The collated recording copies in (c) have been subjected to appropriate scaling and offsetting in order to accommodate the device’s asymmetric threshold voltages, resulting in balanced resistive state SET and RESET. The extracted threshold voltages are identified here as, $V_{eth+} = 1.1V$ and $V_{eth-} = -1.4V$; x-axis for both (c) and (d) is given in S.I. units – each data sample lasts $82\mu s$ (sampling frequency: 12.2 kHz). (e) Fractional resistive state modulation ($\Delta R/R_0$) extrapolated from (d) showcasing significant resistive state modulation occurring only above V_{eth+} and below V_{eth-} while intermediate bias values (noise) leads to no significant change.”*
- Supplementary Figure S5, has been added to support Figure 2 in the main text. The figure clearly depicts the batch processing methodology for processing a single device with respect to an amplified neural trace.

- Supplementary Figure S6, has been added to show the estimation of the extracted threshold voltage and the setting of the noise band limits to extract the significant activity of the MIS system.

b. Is it of mandatory usage?

The characterisation instrument developed by the group is particularly designed to carry out en-masse testing of the RRAM single devices or crossbar arrays. For this work, we have developed bespoke software for “playing-back” the recorded neural-data as distinct biases for the MIS. Clearly, alternative approaches could have been used, i.e. employing a semiconductor characterisation suite and custom interfacing scripts for generating arbitrary neural-like waveforms.

c. How can one run the system with all memristors cell recording synaptic data in parallel? By means of the MTA?

Similarly to our response to referee 1, we envision an APS CMOS imager like architecture for a high-density implementation of the MIS concept (now changed in Fig. 5a), where the MEA system records neural data in parallel from N electrodes at a sampling rate Q, which is then multiplexed to M, higher speed gain and offset stages. This way we may have concurrent processing of M samples in addition to time-multiplexing in order to time-share gain/offset resources, which we thought strikes a good balance between the fully parallel and fully serial approaches. M, of course, and its relation to N is something that can be decided at the design stage, but very importantly, under this scheme samples from all N electrodes are still processed within 1/Q (so we have an ‘effectively parallel’ scheme).

d. If the memristor integrator array is used for integrating simultaneously all synaptic activity (I don't think it is, I would like to see if it is possible), how one deals with multiplexing, half-selected voltages, write disturb from adjacent cells, and all problems that plague 1R memristor array architectures? I understand that in order to make this first and important step one needs a MUX and a DEMUX (Fig. 5) to answer the above question, but one possible goal could be getting rid of all that. Can the authors further elaborate on that?

We presume that the reviewer is asking whether memristor crossbars can somehow be used to parallelise this process fully without the need for this ‘effectively parallel’ architecture. Whilst that is an attractive prospect the key issues with a crossbar implementation are that: i) Crossbars use $2*N$ lines to access N^2 devices. ii) The neural recording data arriving from each channel in the MTA is ‘in principle’ decorrelated. As such, unless we can somehow exploit the correlations across channels in the neural data stream we have no currently obvious way of making N^2 devices react correctly to data from N^2 channels when accessed through only $2*N$ lines. The ‘effectively parallel’ scheme is now shown much more clearly in Fig. 5a, where we have preserved it as our ‘next step of choice’ architecture.

The reviewer is however making an interesting point. Although crossbar offers huge scalability prospects, they suffer from number of challenges in terms of sneak-path issues. If the crossbar configuration is implemented then circuitry similar to those inside the characterisation instrument used in this work to carry out all experiments (Ref: *Serb, A., Berdan, R., Khat, A., Papavassiliou, C. & Prodromakis, T. Live demonstration: A versatile, low-cost platform for testing large ReRAM cross-bar arrays. IEEE Int. Symp. CIRCUITS Syst.* **9**, 4799 (2014)) would be needed in order to deal with sneak path issues. Additionally, the implementation of selectors would also become indispensable for mitigating sneak path issues (the details will depend on the specifics of the memristor technology used). Simultaneously, an array of multiplexers acting effectively as row/column/sub-array decoders, (again the details will vary depending on the specifics of the approach used) will be necessary in order to handle the multiplexing through the higher speed gain/offset stages as described in point (c) above. We can thus combine the ‘effectively parallel’ APS-like structure above with a system similar to our memristor characterisation instrument, where samples from each MTA channel are routed through the instrument to a dedicated memristor at some fixed location within a crossbar.

e. Fig. 3 helps but does not answer all the questions above; fig 5 is a bit more clear, but I am still not sure whether the data flow from cells to memristor was carried out directly as fig. 5 indicates, or if there was an intermediate step to record the data, and then further written by demultiplexing it into the memristor integrated sensor.

We appreciate the reviewer's concern and the figures have now been modified accordingly. The amplified data recorded from the front-end system was saved onto a PC. This is now illustrated in Fig. S3 in the block diagram (PC neural data). These files were acquired and then separately processed through MIS platform as shown in Figure S3. Moreover, all the changes made in the figures have been mentioned in Reviewer's 1 comment, point B specifically. The figure captions, text in the manuscript and methods now clearly indicates that the front-end-system was external to MIS platform.

2) I believe that an overall picture of the system, including the culture cell is important. As mentioned above, the paper is not very clear in saying that fig. 5 is the end goal, and that an intermediate process is carried out (of pre-recording data, and writing data into the memristor array)

We appreciate the concern raised by the reviewer. An additional Fig. S3 has been added to illustrate the overall system indicating the external CMOS chip used for recording the signals from dissected rabbit retinal ganglion cells and the amplified data (in the range of 100's mV) obtained from the front-end system. Moreover, we have changed the concept Figure 2a and the text on Page 2 that now states that the front-end system used in the experiments was 'external' to the system. The block diagram also now illustrates that the front-end system used was external.

On Page 4, paragraph 3 has been changed to: *"The concept introduced in Fig. 2a, when directly interfaced with front-end-circuitry, can be exploited for advancing the present state-of-art in high density neural recording platforms³⁸. The presented concept is amenable for scaling to a multi-channel array level, as illustrated in Fig. 5a, for capturing the activity of neural networks in real-time. We envisage an overall system architecture very similar to standard Active Pixel Sensor (APS) CMOS imagers³⁹. In this hybrid system, data from each of the 'N' pixels in the array arrives as an analogue current from the MEA and is multiplexed onto one of the 'M' on-chip trans-impedance amplifier (TIA) blocks, which are followed by on-chip offset stages..."* to improve the clarity in saying that Fig. 5a remains the end goal and there are number of challenges before this system can be implemented (as mentioned in the same paragraph in the main manuscript). Figure 5 and the figure caption for Fig. 5a has been modified as well: *"Towards array level integration"* which illustrates a conceptual diagram of a hybrid system for conditioning data from 'N' pixels through multiple gain and offset stages (M) instead of single gain and offset stage as initially mentioned.

Clarifying the above points is mandatory prior to publication. Also points 1.a-d help understand that the idea proposed can be scaled up, and that is the key advantage in contrast to existing approaches.

[1] <https://arxiv.org/abs/1507.06832>

[2] Switching dynamics in titanium dioxide memristive devices, Pickett, M.D. et al, *Journal of applied Physics*, 106, 074508 (2009).

[3] Zeitler, R., Fromherz, P. & Zeck, G. Extracellular voltage noise probes the interface between retina and silicon chip. *Appl. Phys. Lett.* 99, 263702 (2011).

[4] Berdan, R. et al. A u-Controller-Based System for Interfacing Selectorless RRAM Crossbar Arrays. *IEEE Trans. Electron Devices* 62, 2190-2196 (2015).

We thank the referee for the provided feedback and we hope that our responses address any concerns raised.

REVIEWERS' COMMENTS:

Reviewer #1 (Remarks to the Author):

The authors have answered my previous comments to my satisfaction and would look forward to this manuscript being published.

Reviewer #2 (Remarks to the Author):

This review refers to the comments made in the first review by Reviewer 2, to the authors' responses to these comments, and to the current version of the paper including revisions made.

Indeed, the authors have carefully considered the comments, related changes and extensions are appreciated. A few remaining issues are found but are of minor significance. Moreover, two new remarks concerning formal issues are provided at the end of this review which can be easily fixed.

Thus, all remaining issues and new comments should thus be satisfactory to the editor. Further re-consideration by the reviewer is not required.

Comment on "Page 2, 2nd paragraph, last two lines", and related changes made:

- Provision of the respective data (mean value and standard deviation) is highly appreciated!
- Whereas content-wise the information presented (mean value and standard deviation is sound), the way of presentation is somewhat lengthy and complicated. Why don't you simply provide mean value (-1.4 V) and 1-sigma or three-sigma standard deviation (0.4 V or 1.2 V)?
- The assumption of a Gaussian distribution is not necessary to calculate a standard deviation. Thus it is proposed to cancel "Assuming a Gaussian distribution ..." and slightly re-phrase the sentence.
- Given the standard deviation of 0.4 V and/or the data of Fig. S2 c) it appears questionable whether the related data may be called "similar" or not, however, this point shall not further be discussed here.

In that context, however, the question may arise for some readers whether that spread is too large for the target application or not. Can you briefly add a short additional comment at a suitable point - maybe in this paragraph or elsewhere - why this spread is not a problem?

- Fig. S2 c): Given the number of 34 devices, the diagram is interpreted that bins of 50 mV width are used, and that in the interval between -2 V and -1.8 V there are two directly neighbouring bins with one device each followed by a number of bins with zero devices. Even in the center of the diagram it seems, that bins with a relatively high number of devices (≥ 4) alternate with bins with zero devices.

Is this correct? Or is this a graphics (or software) related artifact? Please clarify.

- Inset in Fig. S2 c) "-(0.8-1.8)V" may read a little misleading and is not correct in this form

(cf. data). Perhaps better "... from -1.8 V to - 0.8 V ..."?

Comment on "Page 2, last paragraph, upper half", and related changes made:

O.k., response and changes appreciated, no further remarks.

Comment on "Page 2, last but 6th line, ...", and related changes made:

O.k., response and changes appreciated, no further remarks.

Comment on "Page 3, line 8, + several further times, expression 'MTA'":

O.k., response and changes appreciated, no further remarks.

Comment on "Page 4, section 'Towards array-level operation', Fig. 5 and related discussion", and related changes made:

The explanation and extension is appreciated. With respect to the example in the first review (result: 1 GHz BW) we now achieve after the (proposed single) ADC: $1 \text{ GHz} \times \text{number of bits} / \text{data lowering factor}$, i.e. with number of bits = 10 and data lowering factor =200, the output data stream translates into 50 Mbit/s. Whereas such rates as such can be relatively easily handled in CMOS circuits, we approach a region which may be challenging for the envisioned ADC, in particular if we keep the above mentioned resolution at larger BW and larger arrays.

In conclusion it is proposed not to be too strict concerning the number of TIAs and ADCs but replace "a single" by "a single or few ..." or similar.

Comment on "Page 5, last but 5th line, ...", and related changes made:

O.k., response and changes appreciated, no further remarks.

Comment on "Page 6, top", and related changes made:

This question is surely not yet completely covered and decided, but given the paper's target direction this extension is highly appreciated, points into the right direction, and thus should be left as is.

However, the remark "(720 nW-digital input)" remains vague and unclear in this context.

Proposal: leave everything as is but cancel this term in the manuscript.

All further comments made in the first review and related changes:

O.k., response and changes appreciated, no further remarks.

New comments:

1. Fig. S3 b), left: Please increase font size.

2. Figs. 2a), 2b), 5, S3 top right:

A circle symbol is used subdivided by an "X" into four quarters to symbolize an adder with "+" and "-" in respective quarters depending on addition or subtraction of the respective applied input signals. This symbol, however, (without "+" and "-" in part of the quarters) is usually used for multipliers (modulators) in mixers etc.

For addition and subtraction usually circles are used with a "+" inside and a "+" or "-" sign

at arrows pointing onto the circle border and representing the inputs.
The authors may consider to change the figures accordingly to avoid any misunderstanding here.

Reviewer #3 (Remarks to the Author):

I would like to commend the authors for a major restructuring effort to bring clarity to the paper. I have to admit that the paper is rather dense, in the sense that there is a tremendous richness of content that takes a while to be absorbed. I can now see the potential impact of this work, and in summary, I am quite pleased with the modifications.

Now to be more specific:

A) Summary of the key results

The authors were able to significantly improve the overall view, by adding some key figures and related text.

B) Originality and interest: if not novel, please give references

Original and interesting, nothing else to add here.

C) Data & methodology: validity of approach, quality of data, quality of presentation

I had several questions before, but were answered in detail. I am quite pleased.

D) Appropriate use of statistics and treatment of uncertainties

This portion was a significant improvement of the paper. In the previous version it was not quantified, and dispersed in the text by generic words. Now I can see a much more rigorous treatment of the data and explicit mention to statistical analysis.

E) Conclusions: robustness, validity, reliability

As pointed out above, it has been made quite a bit more clear, and I am pleased with the final result.

F) Suggested improvements: experiments, data for possible revision

Not necessary.

G) References: appropriate credit to previous work?

Appropriate credit was given.

H) Clarity and context: lucidity of abstract/summary, appropriateness of abstract, introduction and conclusions

Major changes in clarity from previous version. I am quite please with the overall result and it is ready for publication.

REVIEWERS' COMMENTS:

Reviewer #1 (Remarks to the Author):

The authors have answered my previous comments to my satisfaction and would look forward to this manuscript being published.

We thank the referee for accepting the changes made in the manuscript.

Reviewer #2 (Remarks to the Author):

This review refers to the comments made in the first review by Reviewer 2, to the authors' responses to these comments, and to the current version of the paper including revisions made.

Indeed, the authors have carefully considered the comments, related changes and extensions are appreciated. A few remaining issues are found but are of minor significance. Moreover, two new remarks concerning formal issues are provided at the end of this review which can be easily fixed.

Thus, all remaining issues and new comments should thus be satisfactory to the editor. Further re-consideration by the reviewer is not required.

We thank the referee for acknowledging our efforts we made in restructuring of the manuscript. The remaining issues and the two new remarks are specifically addressed below.

Comment on "Page 2, 2nd paragraph, last two lines", and related changes made:

-Provision of the respective data (mean value and standard deviation) is highly appreciated!

-Whereas content-wise the information presented (mean value and standard deviation is sound), the way of presentation is somewhat lengthy and complicated. Why don't you simply provide mean value (-1.4 V) and 1-sigma or three-sigma standard deviation (0.4 V or 1.2 V)?

-The assumption of a Gaussian distribution is not necessary to calculate a standard deviation. Thus it is proposed to cancel "Assuming a Gaussian distribution ..." and slightly re-phrase the sentence.

-Given the standard deviation of 0.4 V and/or the data of Fig. S2 c) it appears questionable whether the related data may be called "similar" or not, however, this point shall not further be discussed here.

We fully agree with the reviewer. The method for the estimation for effective threshold voltage is indeed a bit lengthy. However, Supplementary Figure 2a was our attempt to provide the readers with the detailed methodology for the estimation of effective threshold voltage of the device, which is a prerequisite in MIS platform for the setting of the gain and offset parameters. We understand from the reader's point of view it can get a bit complicated, therefore we have taken the referee's advice on board and the Supplementary Figure 2a has been omitted. The necessary distributions for the inherent and effective threshold voltage have been now maintained as Supplementary Figure 2a and 2b. The mean and the 3-sigma standard deviation values are mentioned in the Figure caption: *'Inherent and effective threshold of the memristive devices. (a) Range of the 'inherent' threshold voltage (V_{th}/V_{th+}) is approximately ± 0.6 -2.3V as determined by the sample of data obtained from ten devices. (b) Distribution of effective threshold voltage of the TiO_x devices used. This value is used as an approximation for the setting of gain (G) and offset (V_{off}) parameters for the processing of neural recordings before it is used for biasing of the memristive device. The histogram is plotted for a sample of 34 devices. Range of the effective threshold voltages used for the experiment is -0.8V to -1.8V. The*

mean (μ) of the sampled data is equal to $-1.4V$. The one sigma standard deviation values are equal to $(-1V, -1.8V)$.'

In that context, however, the question may arise for some readers whether that spread is too large for the target application or not. Can you briefly add a short additional comment at a suitable point - maybe in this paragraph or elsewhere - why this spread is not a problem?

We appreciate the concern raised by the reviewer and this question is extremely valid given the variability of the employed metal-oxide devices. To further emphasize this point clearly in the main manuscript, the following line has been added on Page 4, paragraph 1: *'Moreover, since the MIS system detects normalised changes in the resistive state, this approach is inherently robust against the devices threshold variability as identified in Supplementary Figure 2b.'*

-Fig. S2 c): Given the number of 34 devices, the diagram is interpreted that bins of 50 mV width are used, and that in the interval between -2 V and -1.8 V there are two directly neighbouring bins with one device each followed by a number of bins with zero devices. Even in the center of the diagram it seems, that bins with a relatively high number of devices (≥ 4) alternate with bins with zero devices.

Is this correct? Or is this a graphics (or software) related artefact? Please clarify.

We apologise for the lack of clarity in the represented data which led to this confusion and was an error related to the plotting of the data in MATLAB. Supplementary Figure 2b, representing the histogram plot for the effective threshold voltage has now been re-plotted where every bin is of 100 mV width.

-Inset in Fig. S2 c) "-(0.8-1.8)V" may read a little misleading and is not correct in this form (cf. data). Perhaps better "... from -1.8 V to -0.8 V ..."?

Amended, as suggested.

-Comment on "Page 2, last paragraph, upper half", and related changes made:

O.k., response and changes appreciated, no further remarks.

-Comment on "Page 2, last but 6th line, ...", and related changes made:

O.k., response and changes appreciated, no further remarks.

-Comment on "Page 3, line 8, + several further times, expression 'MTA'":

O.k., response and changes appreciated, no further remarks.

Comment on "Page 4, section 'Towards array-level operation', Fig. 5 and related discussion", and related changes made:

The explanation and extension is appreciated. With respect to the example in the first review (result: 1 GHz BW) we now achieve after the (proposed single) ADC: 1 GHz x number of bits / data lowering factor, i.e. with number of bits = 10 and data lowering factor = 200, the output data stream translates into 50 Mbit/s. Whereas such rates as such can be relatively easily handled in CMOS circuits, we approach a region which may be challenging for the envisioned ADC, in particular if we keep the above mentioned resolution at larger BW and larger arrays.

In conclusion it is proposed not to be too strict concerning the number of TIAs and ADCs but replace "a single" by "a single or few ..." or similar.

The suggested amendment has been made on Page 5, paragraph 1. The line has been changed to, *'The low frequency at which memristor read-outs are generated (for example 200 times lower data rate*

*vis-à-vis input stream arriving from the MEA if a 'standard scheme is used as described in methods section) allows the MIS system to carry out all measurements through a **single or few**, time-shared TIA feeding into Analogue-to-Digital Converter (ADC).'*

-Comment on "Page 5, last but 5th line, ...", and related changes made:

O.k., response and changes appreciated, no further remarks.

-Comment on "Page 6, top", and related changes made:

This question is surely not yet completely covered and decided, but given the paper's target direction this extension is highly appreciated, points into the right direction, and thus should be left as is.

However, the remark "(720 nW-digital input)" remains vague and unclear in this context. Proposal: leave everything as is but cancel this term in the manuscript.

The term '720 nW-digital input' has now been removed from the main text.

All further comments made in the first review and related changes:

O.k., response and changes appreciated, no further remarks.

New comments:

1. Fig. S3 b), left: Please increase font size.

Modified, as suggested.

2. Figs. 2a), 2b), 5, S3 top right:

A circle symbol is used subdivided by an "X" into four quarters to symbolize an adder with "+" and "-" in respective quarters depending on addition or subtraction of the respective applied input signals. This symbol, however, (without "+" and "-" in part of the quarters) is usually used for multipliers (modulators) in mixers etc.

For addition and subtraction usually circles are used with a "+" inside and a "+" or "-" sign at arrows pointing onto the circle border and representing the inputs.

The authors may consider to change the figures accordingly to avoid any misunderstanding here.

We concur with the referee and indeed this was our mistake in the previous versions of the figure. Figure 2a, 2b, Figure 5 and Supplementary Figure S3 in the main manuscript has now been edited to represent the '+' and '-' symbols correctly to avoid any confusion.

Reviewer #3 (Remarks to the Author):

I would like to commend the authors for a major restructuring effort to bring clarity to the paper. I have to admit that the paper is rather dense, in the sense that there is a tremendous richness of content that takes a while to be absorbed. I can now see the potential impact of this work, and in summary, I am quite pleased with the modifications.

Now to be more specific:

A) Summary of the key results

The authors were able to significantly improve the overall view, by adding some key figures and related text.

B) Originality and interest: if not novel, please give references

Original and interesting, nothing else to add here.

C) Data & methodology: validity of approach, quality of data, quality of presentation

I had several questions before, but were answered in detail. I am quite pleased.

D) Appropriate use of statistics and treatment of uncertainties

This portion was a significant improvement of the paper. In the previous version it was not quantified, and dispersed in the text by generic words. Now I can see a much more rigorous treatment of the data and explicit mention to statistical analysis.

E) Conclusions: robustness, validity, reliability

As pointed out above, it has been made quite a bit more clear, and I am pleased with the final result.

F) Suggested improvements: experiments, data for possible revision

Not necessary.

G) References: appropriate credit to previous work?

Appropriate credit was given.

H) Clarity and context: lucidity of abstract/summary, appropriateness of abstract, introduction and conclusions

Major changes in clarity from previous version. I am quite pleased with the overall result and it is ready for publication.

We thank the reviewer for a very encouraging and positive feedback. Indeed, the comments made in the previous round gave us an opportunity to significantly improve upon the previous version and present our results much more clearly.